# SET-SIZE DEPENDENT COMBINATORIAL BANDITS

## ABSTRACT

This paper introduces and studies a new variant of Combinatorial Multi-Armed Bandits (CMAB), called Set-Size Dependent Combinatorial Multi-Armed Bandits (SSD-CMAB). In SSD-CMAB, each base arm is associated with a set of different reward distributions instead of a single distribution as in CMAB, and the reward distribution of each base arm depends on the set size, i.e., the number of the base arms in the chosen super arm in CMAB. SSD-CMAB involves a much larger exploration set of the super arms than the basic CMAB model. An important property called order preservation exists in SSD-CMAB, i.e. the order of reward means of base arms is independent of set size, which widely exists in real-world applications. We propose the SortUCB algorithm, effectively leveraging the order preservation property to shrink the exploration set. We provide theoretical upper bound of $O\left(\max\left\{\frac{M\delta_L}{\Delta_L}, \frac{L^2}{\Delta_S}\right\}\log(T)\right)$ for SortUCB which outperforms the classic CMAB algorithms with regret $O\left(\frac{ML^2}{\Delta_S}\log(T)\right)$, where $M$ denotes the number of base arms, $L$ denotes the maximum number of base arms in a super arm, $\delta$ and $\Delta$ are related to the gap of arms. We also derive a lower bound which can be informally written as $\Omega\left(\max\left\{\min_{k\in[L]}\left\{\frac{(M-L)\delta_k}{\Delta_k^2}\right\}, \frac{L^2}{\Delta_S}\right\}\log(T)\right)$ showing that SortUCB is partially tight. We conduct numerical experiments, showing the good performance of SortUCB.

## 1 INTRODUCTION

Multi-armed bandit (MAB) (Robbins, 1952; Lai & Robbins, 1985; Auer et al., 2002) is a sequential decision-making problem in which a learner faces a dilemma between exploiting well-observed actions (a.k.a. arms) and exploring new arms that may yield higher rewards. Different from the basic MAB setting, where the learner selects a single arm each time slot, a more general version allows the learner to select a combination of arms (called "base arms") to form a "super arm". The reward of the super arm is the sum of the rewards from all the base arms selected. This generalization is referred to as the Combinatorial Multi-Armed Bandit (CMAB) problem (Gai et al., 2012; Cesa-Bianchi & Lugosi, 2012). Many real-world scenarios can be naturally modeled as CMAB problems. For instance, in the network utility maximization problem (Low & Lapsley, 1999) for shared network resources, where several users share limited resources (e.g., communication links with limited capacity), the objective is to maximize aggregate utility for users. In this case, the utility for each user corresponds to the reward for each base arm and super arms represent combinations of users. Similarly, in online advertising, where each advertisement can be considered as a base arm, and displaying a set of ads together on a website forms a super arm. Due to its practical relevance, a wide range of algorithms have been developed to achieve near-optimal regret in CMAB problem (Kveton et al., 2015; Combes et al., 2015a; Chen et al., 2016b; Wang & Chen, 2018; Merlis & Mannor, 2019).

Despite their generality, most existing CMAB frameworks assume that the unknown distribution of base arms remains fixed, regardless of the properties of the super arm to which they belong. However, in practice, there are scenarios where the distribution of base arms changes when they are pulled in super arms of different sizes, resulting in multiple distributions for each base arm. For example, in utility maximization problems, when selecting more users to share the bandwidth, each user gets a reduced portion, leading to lower utility (Verma & Hanawal, 2020; Fu & Modiano, 2021). Notably, while the reward distribution changes, a better arm still performs better compared to others within super arms of the same size. In the bandit context, this situation can be modeled

as one where the reward mean for each base arm decreases when pulled with a larger number of base arms, meaning each base arm follows multiple distributions. However, the order of base arms remains consistent within super arms of the same size. This property also exists in online advertising where users feel less engaged with a website overloaded with a large number of ads, resulting in lower click-through rates for each advertisement (Wang et al., 2011; Broder et al., 2008). This leads to varying reward means for base arms depending on the size of the super arm. Nevertheless, a high-performing ad still ranks better than others on pages with the same number of ads, even if its distribution changes.

To address the application scenarios described above, we introduce and study the *Set-Size Dependent Combinatorial Multi-Armed Bandit* (SSD-CMAB) problem, with semi-bandit feedback and linear reward function. In SSD-CMAB, combinations of $M$ base arms form the super arm set. Each base arm can be associated with $L$ different distributions, depending on the size of the super arm (up to $L$) that contains it. As a result, there are $ML$ distributions in total for the $M$ base arms. Note that in the CMAB model the reward distribution of a base arm remains the same across all super arms. In contrast, SSD-CMAB models base arm rewards as dependent on the size of the super arm, with each base arm having $L$ different reward distributions. This is the main distinction with CMAB where each base arm has only one fixed distribution (Gai et al., 2012; Combes et al., 2015b), even in non-linear reward settings (Chen et al., 2016b; 2021; Merlis & Mannor, 2019). To the best of our knowledge, previous studies on arms dependent on sets (Chen et al., 2018; Takemori et al., 2020; Fourati et al., 2024) have primarily focused on properties like submodularity, whereas this paper is the first to focus on the order preservation property. See Appendix A for detailed connections and differences between our model and CMAB as well as the literature review.

The parameter space for set-size dependent arms expands significantly, adding considerable complexity to solving the SSD-CMAB problem. Without utilizing the structure of reward for base arms, one would need to independently learn $ML$ distributions (see Appendix B for detailed implementation). However, as demonstrated in previous examples, a common property in SSD-CMAB is *order preservation*: the order of the reward means for base arms remains consistent across super arms of the same size. Traditional CMAB algorithms overlook this property in SSD-CMAB, leading to poor performance as $M$ or $L$ increases. Therefore, an effective algorithm for SSD-CMAB must exploit the order preservation property to reduce the need for learning such a large number of distributions.

**Contributions.** In Section 2, we introduce the SSD-CMAB problem. We propose the SortUCB algorithm afterwards which leverages the order preservation property of super arms with the same number of base arms. The algorithm first learns the order of base arms in fixed-size super arms, then identifies and retains a subset of super arms likely to be optimal, excluding suboptimal ones with high probability. Finally, it applies a UCB-based strategy to select super arms. By utilizing the order preservation property, SortUCB minimizes exploration on suboptimal super arms, allowing it to focus on those more likely to be optimal.

SortUCB achieves a regret upper bound of $O\left(\max\left\{\frac{M\delta_{L,\max}}{\Delta_{L,\min}^2}, \frac{L^2}{\Delta_{S,\min}}\right\}\log(T)\right)$, where $M$ is the total number of base arms, $L$ is the maximum size of a super arm, $\Delta_{S,\min}$ is the minimum gap among super arms, and $\Delta_{L,\min}$ is the minimum gap among the top $L$ base arms in size-$L$ super arms. In comparison, applying existing CMAB algorithms yields a regret bound of $O\left(\frac{ML^2}{\Delta_{S,\min}}\log(T)\right)$, which depends only on $\Delta_{S,\min}$. Our bound, however, accounts for $\Delta_{L,\min}$ and introduces $\delta_{L,\max}$, the maximum regret when pulling the top $L$ base arms, which is less than 1. Since $L$ can be at most $M$, the $ML^2$ term in existing bounds grows rapidly with large $M$. By decomposing $ML^2$ into $L^2$ and $M$, our bound ensures better performance, particularly when the number of base arms is exponentially large.

We derive a lower bound for the SSD-CMAB problem, informally expressed as $\Omega\left(\max\left\{\sum_{i=\ell^*+1}^{M}\min_{\ell\in[L]}\left\{\frac{\delta_{i(\ell)}-s_{i(\ell)}}{\Delta_{i(\ell),\ell^*(\ell)}^2}\right\}, \frac{L^2}{\Delta_{S,\min}}\right\}\log(T)\right)$, where $\left(\delta_{i(\ell)} - s_{i(\ell)}\right)$ represents the regret from pulling base arm $i$ in a super arm of size $\ell$. This near-optimal lower bound aligns closely with our regret upper bound. Specifically, the first term in the lower bound corresponds to the first term in the upper bound, indicating tightness. The second term in the lower bound, which aggregates the minimum $\left(\frac{\delta_{i(\ell)}-s_{i(\ell)}}{\Delta_{i(\ell),\ell^*(\ell)}^2}\right)$ for all $\ell \in [L]$ across suboptimal base arms,

aligns with the second term in the upper bound when the index for the minimum is $L$. Additionally, the $\log(T)$ factor in our upper bound is optimal, demonstrating that SortUCB achieves logarithmic regret growth as $T$ increases.

In Section 5, we extend our model to the Set-Dependent Combinatorial Multi-Armed Bandits (SD-CMAB), where the mean reward of each base arm depends on the specific set of arms rather than just the set size. This extension introduces additional complexity, as not all combinations of base arms form valid super arms, making the reward structure more challenging to learn. To address this, we propose the SortUCB-SD algorithm and derive its regret upper bound. Finally, in Section 6, we present numerical experiments showcasing the effectiveness of our approach.

## 2 THE SSD-CMAB PROBLEM

This section introduces the Set-Size Dependent Combinatorial Multi-Armed Bandit (SSD-CMAB) problem which defines size-dependent rewards and feedback for base arms. We begin with a brief explanation of the notations used in this paper.

**Notations.** Throughout the paper, we use $[n] := \{1, 2, ..., n\}$ to denote the set of indexes to simplify notations. For two vectors with the same size $\boldsymbol{\mu} = (\mu_1, \ldots, \mu_n)$ and $\boldsymbol{\nu} = (\nu_1, \ldots, \nu_n)$, we define $\boldsymbol{\mu} \succeq \boldsymbol{\nu}$ when $\mu_i \geq \nu_i$ holds for all $i \in [n]$. Notation $\preceq$ is defined in a similar way by replacing $\geq$ with $\leq$.

An SSD-CMAB problem instance $\nu$ involves $M$ *base arms*, denoted by set $[M]$. Consider a time horizon of length $T$, the player can select a subset of base arms at each time slot as a *super arm*. Let $\mathcal{S}$ denote the set of all possible subsets of base arms $[M]$ whose cardinality is no more than $L \in \mathbb{Z}_+$, i.e., $\mathcal{S} := \{S \subseteq [M] : |S| \leq L\}$ (meaning a super arm consists of at most $L$ base arms).

Unlike the classical stochastic CMAB problem, where each base arm's reward follows a fixed i.i.d. distribution, the base arm rewards in SSD-CMAB are *set-size dependent*. Specifically, for any base arm $i$, when it is pulled as part of a super arm $S \in \mathcal{S}$ with size $\ell = |S|$, its reward follows a distribution dependent on $\ell$, denoted as $P_{i(\ell)}$. For simplicity, we use $\ell_S$ to represent the size of super arm $S$. Since a super arm consists of at most $L$ base arms, each base arm $i$ has at most $L$ possible reward distributions. Without loss of generality, we assume base arm rewards are $[0, 1]$-valued. Let $\mu_{i(\ell)}$ denote the expected reward for arm $i$ under $P_{i(\ell)}$, and $\boldsymbol{\mu}_i = (\mu_{i(1)}, \mu_{i(2)}, \ldots, \mu_{i(L)})$ represent the vector of mean rewards for arm $i$ across different set sizes. This dependency on super arm size defines the "Set-Size Dependent" nature of the problem.

We denote by $N_{i(\ell),t}$ the number of times base arm $i$ has been pulled under distribution $P_{i(\ell)}$ up to time slot $t$, and by $X_{i(\ell),N_{i(\ell),t}}$ the outcome of base arm $i$ at time slot $t$ under the same distribution. Let $S_t$ and $R(S_t)$ represent the super arm chosen and its corresponding reward at the $t$-th time slot, respectively, with the expected reward denoted as $r(S_t) = \mathbb{E}[R(S_t)]$. We consider a linear reward function where $R(S_t) = \sum_{i \in S_t} X_{i(\ell_{S_t}),N_{i(\ell),t}}$, so $\mathbb{E}[X_{i(\ell),N_{i(\ell),t}}] = \mu_{i(\ell)}$. The average reward of base arm $i$ over the first $t$ time slots within super arms of size $\ell$ is denoted by $\hat{\mu}_{i(\ell),N_{i(\ell),t}} = \sum_{s=1}^{N_{i(\ell),t}} X_{i(\ell),s}/N_{i(\ell),t}$. In this paper we consider the semi-bandit feedback, where the learner selects a super arm $S \in \mathcal{S}$ each time slot and observes the rewards for all base arms in $S$. As mentioned in the Introduction, the order preservation property exists in the SSD-CMAB model. We formally introduce it as follows:

**Order preservation.** For any class $\ell \in [L]$, the order of reward expectations is fixed across different base arms. That is, $\mu_{i(\ell)} \leq \mu_{j(\ell)}$ if and only if $\mu_{i(\ell')} \leq \mu_{j(\ell')}$, where $i, j \in [M]$, $\ell, \ell' \in [L]$.

An SSD-CMAB algorithm $\pi$ selects one super arm $S$ to play each time slot according to the previous information. The objective of $\pi$ is to maximize the cumulative expected reward in $T$ time slots. We use $S^* = \arg\max_{S \in \mathcal{S}} r(S)$ to denote the optimal super arm. In order to show the performance between an algorithm $\pi$ and the optimal policy (i.e. always pull the optimal arm) on the instance $\nu$, we need a quantity *'Regret'* defined as

$$\text{Reg}_T(\pi, \nu) = T \cdot r(S^*) - \mathbb{E}\left[\sum_{t=1}^{T} r(S_t)\right]. \tag{1}$$

Thus, the objective of algorithm $\pi$ is to minimize $\text{Reg}_T(\pi, \nu)$.

## 3 ALGORITHM: SORTING UPPER CONFIDENCE BOUND

In this section, we introduce details of our algorithm Sorting Upper Confidence Bound (SortUCB).

Compared to traditional CMAB problems, SSD-CMAB is faced with a larger challenge in handling the super arms which involve more reward distributions with the same number of base arms. Consider an SSD-CMAB problem with a maximum super arm size of $L$. As a result, the parameters or reward means to learn expands by a factor of $L$ compared to the CMAB setting with a single distribution associated to each base arm. Those make traditional CMAB algorithms which directly learn the reward means of super arms fail to maintain efficient in handling massively large amount of parameters. Those challenges urge us to leverage a structured exploration strategy which guides the algorithm to assign pulls to the super arms from which the algorithms can obtain more information on the structure or order specifically of the base arms. From the analysis in Section 4, the above strategy can efficiently lower down the pulls of arms whose reward means subject to a particular nature, i.e. the order preservation property. Hence, the learning algorithm for SSD-CMAB, compared to those for the traditional CMAB problems, contains additional Elimination Phase and Sorting Phase where the algorithm needs to learn the structure of reward means and eliminate suboptimal super arms according to the learned structure. However, the above strategy introduces another source of *exploration-exploitation* dilemma between assigning pulls to learn the structure to eliminate super arms or directly applying classic bandit learning algorithms to learn the best super arm. The above dilemma results in the second challenge of SSD-CMAB. To address the above two challenges, we present our SortUCB algorithm in Algorithm 1 which effectively leverages the order preservation property and learns the structure of reward distributions in an appropriate way.

As mentioned above, the core idea behind the structured exploration strategy in the proposed algorithm is to leverage the order preservation property to avoid exploring unnecessary super arms. Specifically, by pulling certain super arms, the algorithm can learn the order among some base arms. Since any combination of base arms can form a super arm, the super arm set is exponentially large. However, with the learned order, the algorithm manage to identify some super arms as suboptimal because the base arms they include have smaller reward means than others according to the learned order. For example, if the algorithm figures out that base arm 1 is better than base arm 2, there is no need to pull super arms such as $\{2, 3\}$ or $\{2, 4\}$, as these super arms are worse than $\{1, 3\}$ and $\{1, 4\}$. This means that if some particular order is learned during earlier samplings to some degree (correspond to the Sorting Phase in Algorithm 1), the order preservation property allows us to reduce pulling all the super arms that contain a base arm which is likely to perform poorly. The analysis later on shows that the above strategy significantly narrows down the set of super arms that the algorithm needs to explore. We introduce the details of implementing the algorithm below.

---

**Algorithm 1** Sorting Upper Confidence Bound

---

1: **Initialization:** $\mathcal{B} \leftarrow [M]$
2: \\Elimination Phase                       $\triangleright$ Learn the best $L$ base arms
3: **while** $|\mathcal{B}| > L$ **do**
4:     Pull the super arm consisting $L$ smallest $N_{i(L),t}$ base arms (uniform pull)
5:     Update $\hat{\mu}_{i(L),t}, N_{i(L),t}$ and $t$
6:     Delete all the base arms satisfying (2) for $L$ different base arms $j_1, \ldots, j_L$
7: **end while**
8: $\mathcal{R} \leftarrow \mathcal{B}$
   \\Sorting Phase                         $\triangleright$ Sort the best $L$ base arms
9: **while** $|\mathcal{B}| \geq 2$ **do**
10:     Pull the super arm $\mathcal{R}$, and update $\hat{\mu}_{i(L),t}, N_{i(L),t}$ and $t$
11:     Delete any base arm $i$ satisfying (2) for all $j \in \mathcal{B} \setminus i$, and set the order of arm $i$ to $|\mathcal{B}| + 1$
12: **end while**
13: Set $\mathcal{A}$ as (3), $\hat{\mu}_{i(\ell),N_{i(\ell),t}} \leftarrow 0$ for all possible $i$ and $\ell$, $N_{S,t} \leftarrow 0$ for super arms $S$ in $\mathcal{A}$
   \\UCB Phase                    $\triangleright$ Using UCB to select a super arm each slot
14: **while** $t \leq T$ **do**
15:     Pull super arm with the highest (4) for super arms $S \in \mathcal{A}$
16:     Update $N_{S,t}, \hat{\mu}_{i(\ell),t}$ and $t$.
17: **end while**

---

The algorithm begins with exploring the order of base arms, which includes the "Elimination Phase" and "Sorting Phase". We use $\mathcal{B} = [M]$ to represent the set of all base arms in our algorithm. As stated in the second challenge, the algorithm maintains a fixed super arm size during the exploration phase, uniformly pulling super arms with the largest number of base arms (i.e., super arms of size $L$) to gather as much information as possible.

Since each super arm contains no more than $L$ base arms, it is unnecessary to precisely estimate those that are not among the best $L$ base arms, as attempting to learn about these arms can lead to substantial regret. Thus the algorithm adjusts its policies to learn on different base arms between the first two phases. Initially in the Elimination Phase, the algorithm focuses on identifying the $(M - L)$ base arms that are not among the best $L$ and removes them from $\mathcal{B}$, rather than determining their exact order. Specifically, for a base arm $i \in \mathcal{B}$, if there exist at least $L$ base arms $j_k$ $(k \in [L])$ whose lower confidence bound exceeds the upper confidence bound of $i$, i.e.,

$$\hat{\mu}_{i(L),t} + \sqrt{\frac{2\log(T)}{N_{i(L),t}}} < \hat{\mu}_{j_k(L),t} - \sqrt{\frac{2\log(T)}{N_{j_k(L),t}}}, \tag{2}$$

for $L$ different base arms $j_1, j_2, \ldots, j_L \in \mathcal{B}$, then we remove base arm $i$ from $\mathcal{B}$, as it is suboptimal with high probability regarding these $L$ base arms. This process continues until $\mathcal{B}$ contains no more than $L$ base arms, which means it now with high probability holds the best $L$ base arms.

In the Sorting Phase, the algorithm shifts to determining the exact order of the remaining $L$ base arms, as their ranking is essential for exploiting the order preservation property. We define the super arm $\mathcal{R} = \mathcal{B}$, which includes the top $L$ base arms. The algorithm continues pulling $\mathcal{R}$ and removes any base arm $i$ from $\mathcal{B}$ that satisfies the condition in (2) for all $j_k \in \mathcal{B} \setminus \{i\}$, thus learning that the rank of $i$ is $|\mathcal{B}| + 1$. This procedure concludes when only one base arm remains in $\mathcal{B}$, which is identified as the best base arm with high probability.

Afterwards, we can use the order preservation property to remove a large number of suboptimal super arms. We use $\mathcal{A}$ to denote the set of super arms containing the top $\ell$ base arms ($\ell \in [L]$) identified in the previous phase. That is,

$$\mathcal{A} = \{\{1, \ldots, \ell\} \,|\, \ell \in [L]\}. \tag{3}$$

With high probability, the optimal super arm is within $\mathcal{A}$, as the best super arm for each size belongs to this set, and the overall optimal super arm must be one of them.

The remainder of the algorithm (UCB Phase) focuses solely on exploitation within this set. To proceed, we reset all estimates of the base arms $\hat{\mu}_{i(\ell)}$, allowing us to use an extended version of the UCB algorithm. This version treats each super arm as a single arm and tracks the number of times each super arm has been selected. We use $N_{S,t}$, instead of $N_{i(\ell),t}$, to denote the number of times super arm $S$ has been selected by time $t$. In this phase, the algorithm pulls the super arm $S \in \mathcal{A}$ with the highest value of

$$\left(\sum_{i \in S} \hat{\mu}_{i(\ell_S), N_{i(\ell_S),t}}\right) + \sqrt{\frac{2|S|\log(T)}{N_{S,t}}}. \tag{4}$$

**Implementation of Algorithm 1.** Algorithm 1 can be implemented with a computational complexity of at most $O(M\log(M))$ per time slot. Specifically, the first two phases involve sorting the $M$ base arms and eliminating suboptimal ones, which can be performed with complexities of $O(M\log(M))$ and $O(M)$ per time slot, respectively. In the third phase, the algorithm applies a UCB-like strategy on $|\mathcal{A}| = L$ super arms, which requires $O(L)$ complexity per time slot. A detailed explanation of the computational complexity is provided in Appendix C.

## 4 Theoretical Analysis

### 4.1 Instance Dependent Upper Bound

In this subsection we give our theoretical results, including the instance dependent upper bound for Algorithm 1 and the instance dependent lower bound for SSD-CMAB problem.

**Theorem 1.** *For any SSD-CMAB instance $\nu$, the regret of SortUCB is bounded as:*

$$\text{Reg}_T(\text{SortUCB}, \nu) \leq O\left(\left(\frac{M\delta_{L,\max}}{\Delta_{L,\min}^2} + \frac{L^2}{\Delta_{S,\min}}\right)\log(T)\right). \tag{5}$$

*Here $\delta_{L,\max} = \max_{i \in [M]}\{\delta_{i(L)}\}$, where $\delta_{i(\ell)} = r(S^*)/\ell - \mu_{i(\ell)}$ for $i \in [M]$ and $\ell \in [L]$, which varies in $[-1, 1]$ (note $\delta_{i(\ell)}$ can be negative when $i < \ell$ but term $\delta_{L,\max}$ keeps positive). $\Delta_{L,\min} = \min_{i \in [L]} \Delta_{i(L),(i+1)(L)}$ denotes the minimum gap of reward mean between any two adjacent base arms in the super arm $\{1, \ldots, L\}$.*

**Remark 1** (Intuitive Explanation for Regret). *The first term $O\left((M\delta_{L,\max}/\Delta_{L,\min}^2)\log(T)\right)$ of the regret upper bound in Equation (5) is introduced by the Elimination and Sorting Phases, where the algorithm eliminates the $(M - L)$ worst base arms and learns the order of the first $L$ base arms, while the second term $O\left((L^2/\Delta_{S,\min})\log(T)\right)$ in Equation (5) comes from the UCB Phase, exploring the set $\mathcal{A}$ of possible optimal super arms composed by $O(L^2)$ base arms, leading to a regret cost similar to that of the standard UCB algorithm.*

**Remark 2** (Comparison with CMAB Results). *While our SSD-CMAB model could be reduced to a traditional CMAB with linear reward function, the state-of-the-art result for CMAB is $O\left(\frac{ML^2}{\Delta_{S,\min}}\log(T)\right)$ by the CombUCB1 algorithm (Kveton et al., 2015). This bound is much worse than that of our SortUCB algorithm, where the $ML^2$ factor of the CombUCB1 is improved to $(M + L^2)$.*

*The only loose part in the upper bound compared to the lower bound in Theorem 2 of SortUCB is the factor $\Delta_{L,\min}^2$ in the denominator of the first term. In most real-life cases, the gap $\Delta_{L,\min}$ among the $L$ base arms is not that small as $\Delta_{S,\min}$, and hence SortUCB performs well in practice (see the experiments in Section 6).*

### 4.2 INSTANCE DEPENDENT LOWER BOUND

For the instance dependent lower bound, we consider an SSD-CMAB instance $\mathcal{E} = \mathcal{M}_1 \times \cdots \times \mathcal{M}_L$, where $\mathcal{M}_\ell$ ($\ell \in [L]$) is a set of distribution vectors $\mathcal{P} = (P_1, \ldots, P_M)$ satisfying $\mu(P_1) \geq \mu(P_2) \geq \cdots \geq \mu(P_M)$, denoting the mean for all the $M$ base arms in super arms with size $\ell$. The theoretical result of the lower bound depends on two extra definitions. We formally introduce them as below.

**Definition 1.** *A policy $\pi$ is called consistent over a class of bandits $\mathcal{E}$ when for all $\nu \in \mathcal{E}$ and $p > 0$, it holds that*

$$\lim_{T \to \infty} \frac{\text{Reg}_T(\pi, \nu)}{T^p} = 0.$$

*We use $\Pi_{cons}(\mathcal{E})$ to denote consistent policies over $\mathcal{E}$.*

**Definition 2.** *Let $\mathcal{M}$ be a set of distributions with finite means, and let $\mu : \mathcal{M} \to \mathbb{R}$ be the function that maps $P \in \mathcal{M}$ to its mean. Let $\mu^* \in \mathbb{R}$ and $P \in \mathcal{M}$ such that $\mu(P) < \mu^*$. We define:*

$$d_{\inf}(P, \mu^*, \mathcal{M}) = \inf_{P' \in \mathcal{M}}\{D(P, P') : \mu(P') > \mu^*\}.$$

Suppose $\pi \in \Pi_{cons}(\mathcal{E})$ is a consistent policy over $\mathcal{E}$. The lower bound is indeed to calculate $R_T(\pi, \nu)/\log(T)$ for all possible $\nu \in \mathcal{E}$ when $T$ tends to infinity.

**Theorem 2.** *For all $\nu = (\mathcal{P}_\ell)_{\ell=1}^L \in \mathcal{E}$, it holds that $\lim_{T \to \infty} \inf \frac{\text{Reg}_T(\pi, \nu)}{\log(T)} \geq$*

$$\max\left\{\sum_{i=\ell^*+1}^M \min_{\ell \in [L]}\left\{\frac{\delta_{i(\ell)} - s_{i(\ell)}}{d_{\inf}(P_{i(\ell)}, \mu_{\ell^*(\ell)}, \mathcal{M}_\ell)}\right\}, \sum_{\ell:\ell \neq \ell^*} \frac{\Delta_{\{1,\ldots,\ell\}}}{d_{\inf}(\sum_{j=1}^\ell P_{j(\ell)}, r(S^*), \mathcal{M}_\ell)}\right\},$$

*where $\Delta_{\{1,\ldots,\ell\}} = r(S^*) - \sum_{j=1}^\ell \mu_{j(\ell)}$ denotes the gap of reward mean for best $\ell$ base arms with super arm size $\ell$, and $s_{i(\ell)} = r(\{1, \ldots, \ell\})/\ell - \mu_{(\min\{\ell,i\})(\ell)}$. $P_{j(\ell)}$ indicates the distribution for the $j$-th term in vector $\mathcal{P}_\ell$, $d_{\inf}(P_{i(\ell)}, \mu_{\ell^*(\ell)}, \mathcal{M}_\ell) = \inf_{\mathcal{P}' \in \mathcal{M}_\ell}\{D(P_{i(\ell)}, P'_{i(\ell)}) : \mu(P'_{i(\ell)}) > \mu_{\ell^*(\ell)}\}$, and*

$$d_{\inf}\left(\sum_{j=1}^\ell P_{j(\ell)}, r(S^*), \mathcal{M}_\ell\right) = \inf_{\mathcal{P}' \in \mathcal{M}_\ell}\left\{D\left(\sum_{j=1}^\ell P_{j(\ell)}, \sum_{j=1}^\ell P'_{j(\ell)}\right) : \sum_{j=1}^\ell \mu(P'_{j(\ell)}) > r(S^*)\right\}.$$

**Remark 3.** *In Theorem 2, we use the KL-Divergence for any two i.i.d distributions to present the lower bound. In order to compare it to the upper bound in Theorem 1, we consider the case where each $P_{i(\ell)}$ follows a normal distribution $\mathcal{N}(\mu_{i(\ell)}, 1)$ for $i \in [M]$, $\ell \in [L]$. Then, $d_{\inf}(\sum_{j=1}^{\ell} P_{j(\ell)}, r(S^*), \mathcal{M}_\ell)$ equals to $\Delta_{\{1,\ldots,\ell\}}^2/\ell$. Hence, the first term in Theorem 2 can be rewritten as $\sum_{i=\ell^*+1}^{M} \min_{\ell \in [L]} \left\{ \frac{\delta_{i(\ell)} - s_{i(\ell)}}{\Delta_{i(\ell),\ell^*(\ell)}^2} \right\}$, while the second term is $\sum_{\ell:\ell \neq \ell^*} \frac{\ell}{\Delta_{\{1,\ldots,\ell\}}}$.*

**Remark 4** (Comparison Between Upper and Lower Bounds)**.** *Both the lower bound in Theorem 2 and the upper bound in Theorem 1 have two terms. With their second terms matching, their key distinction is the difference between the first terms which leads to their partially matching. The first term of the lower bound considers the minimum for $(\delta_{i(\ell)} - s_{i(\ell)})/\Delta_{i(\ell),\ell^*(\ell)}^2$ across all $\ell \in [L]$ for each base arm $i$ where the numerator $(\delta_{i(\ell)} - s_{i(\ell)})$ represents the regret incurred by pulling base arm $i$ within super arm $\{1, \ldots, \ell\}$. However, this first term of the upper bound in Theorem 1 is restricted to one $L$, instead of the minimum across $[L]$. Additionally, the size of the summation range of the first term in the lower bound is $(M - \ell^*)$, different from the $M$ in the upper bound. Note that if the minimum in the first term of Theorem 2 across $\ell \in [L]$ consistently falls on $L$, and $\ell^*$ is not approximate to $M$, then the upper and lower bounds align.*

### 4.3 SKETCH OF PROOF

**Proof Sketch (Theorem 1).** We defer the full proof to Appendix D and Appendix E and discuss the sketch proof below. SortUCB has three different phases and the regret of SortUCB could decomposed into three parts: Elimination Phase part, Sorting Phase and UCB Phase part. Therefore, we first give lemmas about the regret produced by three phases below.

**Lemma 1.** *For any SSD-CMAB instance $\nu$, the total regret produced in the Elimination Phase and the Sorting Phase in Algorithm 1 on instance $\nu$, denoted as $\text{Reg}_T(1, \nu)$, is bounded as:*

$$\sum_{i=L+1}^{M} \frac{32 \log(T)}{\Delta_{i(L),L(L)}^2} \delta_{i(L)} + \frac{32 L \log(T)}{\min_{j \in [L]} \left\{ \Delta_{j(L),(j+1)(L)}^2 \right\}} \delta_{L(L)} + \left( 2 + \frac{2ML}{T^2} \right) \sum_{i=1}^{M} \delta_i. \quad (6)$$

The first term in Lemma 1 arises from the Elimination Phase, where each base arm's order is determined by ensuring that condition (2) holds. It can be shown that for each base arm $i$ from $L+1$ to $M$, the inequality $N_{i(L),T} \leq \frac{32 \log(T)}{\Delta_{i(L),L(L)}^2}$ is satisfied. The second term originates from the Sorting Phase, during which the algorithm pulls the first $L$ base arms together. It can be verified that the orders of these base arms can be learned within at most $\frac{32 \log(T)}{\min_{j \in [L]} \{\Delta_{j(L),(j+1)(L)}^2\}}$ time slots. By summing these two components, Lemma 1 is derived. These two terms are combined because of their similar forms, resulting in a total bound of $O\left( \frac{M \delta_{L,\max}}{\Delta_{L,\min}^2} \right)$.

**Lemma 2.** *For any SSD-CMAB instance $\nu$, the regret produced in the UCB Phase in Algorithm 1 on instance $\nu$, denoted as $\text{Reg}_T(2, \nu)$, is bounded as:*

$$\sum_{S \in \mathcal{A}, S \neq S^*} \left( 3\Delta_S + \frac{8|S| \log(T)}{\Delta_S} \right) + \frac{ML}{T^2} \sum_{S \in \mathcal{A}} \Delta_S. \quad (7)$$

Lemma 2 gives the regret from the UCB Phase after line 14 in our algorithm. Here we treat each super arm as a single arm, and could obtain Lemma 2 by using standard analysis of UCB. Finally, Theorem 1 can be proved by summing $\text{Reg}_T(1, \nu)$ and $\text{Reg}_T(2, \nu)$ up.

**Proof Sketch (Theorem 2).** Note that the instance dependent lower bound in Theorem 2 for SSD-CMAB problem also includes two parts. In fact, this is due to the two different ways we use to prove the lower bound, leading to the lower bound being the maximum of the two results. Below we give these two parts as two lemmas in turn, showing the proof sketch. Also we suppose $\pi \in \Pi_{cons}(\mathcal{E})$ is a consistent policy over $\mathcal{E}$. We begin with proving the first term in Theorem 2.

**Lemma 3.** *For all $\nu = (\mathcal{P}_\ell)_{\ell=1}^{L} \in \mathcal{E}$, it holds that*

$$\lim_{T \to \infty} \inf \frac{R_T(\pi, \nu)}{\log(T)} \geq \sum_{i=\ell^*+1}^{M} \min_{\ell \in [L]} \left\{ \frac{\Delta_{\{1,\ldots,\ell\}}/\ell + \max\{\mu_{\ell(\ell)} - \mu_{i(\ell)}, 0\}}{d_{\inf}(P_{i(\ell)}, \mu_{\ell^*(\ell)}, \mathcal{M}_\ell)} \right\}.$$

In order to prove Lemma 3, we just need to bound $N_{i(\ell),T}$ for each base arm $i$ and super arm size $\ell$. We consider another SSD-CMAB instance $\nu' = (P'_{j(\ell)})_{j\in[M],\ell\in[L]} \in \mathcal{E}$. For base arm $j \neq i$, let $P_{j(\ell)} = P'_{j(\ell)}$, and let $P'_{i(\ell)}$ satisfy both $\mathrm{D}(P_{i(\ell)}, P'_{i(\ell)}) \leq d_{\inf}(P_{i(\ell)}, \mu_{\ell^*(\ell)}, \mathcal{M}_\ell) + \varepsilon$ and $\mu(P_{\ell^*(\ell)}) < \mu(P'_{i(\ell)}) < \mu(P_{(\ell^*-1)(\ell)})$ for each $\ell \in [L]$. Using the Bretagnolle-Huber inequality (Lemma 15 in Appendix E), we can derive a weighted lower bound for all $N_{i(\ell),T}$,

$$\lim_{T\to\infty} \frac{\sum_{\ell=1}^{L} \mathbb{E}_{\nu\pi}[N_{i(\ell),T}](d_{\inf}(P_{i(\ell)}, \mu_{\ell^*(\ell)}, \mathcal{M}_\ell) + \varepsilon)}{\log(T)} \geq 1. \tag{8}$$

Rearranging the weight for each $N_{i(\ell),T}$ where $\ell \in [L]$ and summing that for all base arm $i \in \{\ell^* + 1, \ldots, M\}$, we obtain Lemma 3. We furtherly discuss the second term in Theorem 2.

**Lemma 4.** *For all $\nu = (\mathcal{P}_\ell)_{\ell=1}^{L} \in \mathcal{E}$, it holds that*

$$\lim_{T\to\infty} \inf \frac{R_T(\pi, \nu)}{\log(T)} \geq \sum_{\ell:\ell\neq\ell^*} \frac{\Delta_{\{1,\ldots,\ell\}}}{d_{\inf}(\sum_{j=1}^{\ell} P_{j(\ell)}, \mu_{S^*}, \mathcal{M}_\ell)}.$$

Lemma 4 is proved through a mapping technique. Specifically, we consider a map from policy $\pi$ to $\pi'$, where at time slot $t$, the super arm selected by $\pi'$ has the same size as that selected by $\pi$, but $\pi'$ always chooses the optimal super arm of that size. In other words, if $\pi$ selects a super arm of size $\ell$ at time $t$, then $\pi'$ selects the super arm consisting of the best $\ell$ base arms. This mapping restricts the action space to a set of totally $L$ super arms, where each super arm follows the distribution of $(\sum_{j=1}^{\ell} P_{j(\ell)})$. Applying standard techniques for lower bound analysis, we then derive Lemma 4. Combining these Lemma 3 and 4, we can derive the result as shown in Theorem 2.

## 5 EXTENSION TO SET DEPENDENT COMBINATORIAL BANDITS

In this section, we generalize the setting to cover applications where the base arm reward distributions may be different even in the super arms with the same set-size, and the set of feasible super arms can be arbitrary, which could be subjective to any combinatorial constraints (e.g., matroids, paths, matchings), rather than super-arms whose cardinality is less or equal to $L$. We call the model *Set Dependent Combinatorial Multi-Armed Bandit* (SD-CMAB for short). In SD-CMAB, we consider $M$ base arms with a feasible super arm set $\mathcal{S} \subseteq \{S \subseteq [M] : |S| \leq L\}$ as the action set, rather than $\mathcal{S}' := \{S \subseteq [M] : |S| \leq L\}$ in SSD-CMAB. We define a key concept termed as *class*, where $\mathcal{S}$ can be partitioned into $K$ classes, denoted by $\{\mathcal{S}_1, \ldots, \mathcal{S}_K\}$, $K$ indicates the total number of classes. That is, $\bigcup_{i\in[K]} \mathcal{S}_i = \mathcal{S}$, $\mathcal{S}_k \cap \mathcal{S}_{k'} = \emptyset$ for any two different $k, k' \in [K]$, where $\emptyset$ denotes an empty set. Each base arm $i$ is assigned $K$ different distributions $\mathbb{P}_{i(k)}$ for $k \in [K]$. And the reward of base arm $i$ follows distribution $\mathbb{P}_{i(k)}$ when pulled in super arms from class $k$. We use $\boldsymbol{\mu}_i = (\mu_{i(1)}, \ldots, \mu_{i(K)})$ to denote the vector of expected reward for base arm $i$ in super arms within different classes. The order preservation property also holds for SD-CMAB. The rest of the settings (e.g., base/super arm reward, feedback) are the same as Section 2. As defined in (1), the objective is to find an algorithm $\pi$ to minimize the cumulative regret $\mathrm{Reg}_T(\pi, \nu)$ on bandit instance $\nu$. To this end, we can see that SSD-CMAB is in fact the special case of SD-CMAB when $\mathcal{S}_\ell = \{S \in [M] : |S| = \ell\}$.

We propose an algorithm which is an extension version of SortUCB, called Sorting Upper Confidence Bound - Set Dependent (SortUCB-SD). Similar to SortUCB, the core idea in SortUCB-SD is to leverage the order preservation property to learn the reward distribution structure for each base arm within different super arms across various classes. After eliminating a large number of suboptimal super arms, exploration is conducted on the remaining set of super arms. However, unlike SortUCB, SD-CMAB lacks the desirable property where the distribution for each base arm only changes when it is pulled in super arms of different sizes, and certain combinations of base arms cannot form a valid super arm for selection. Therefore, the algorithm relies on an *$(n_1, n_2)$-efficiency Oracle* (explained in Appendix F) to guide it in determining which orders of base arms to focus on learning (denoted by $\mathcal{B}_h$), and which super arms should be pulled to achieve this learning (denoted by $\mathcal{R}_h$). Due to space limit, we postpone the detailed algorithm with an intuitive example in Appendix F. Here we propose Theorem 3 to show the upper bound for Algorithm 3.

**Theorem 3.** *For any* SD-CMAB *instance $\nu$, the regret of* SortUCB-SD *is bounded as:*

$$\text{Reg}_T(\text{SortUCB-SD}, \nu) \leq O\left(\sum_{h=1}^{H}\left(\frac{32\log(T)}{\Delta_{\mathcal{B}_h,\min}^2}\sum_{S\in\mathcal{R}_h}\Delta_S\right) + \sum_{S\in\mathcal{G},S\neq S^*}\frac{8|S|\log(T)}{\Delta_S}\right).$$

**Remark 5.** *Given that $\mathcal{B}_h$ is $(\alpha_h, \beta_h)$-efficient for each $h \in [H]$ where $\alpha_h$ and $\beta_h$ are inputs of Algorithm 3, we have $|\mathcal{R}_h| \leq \alpha_h$ and $\mathcal{G} \leq |\mathcal{S}| - \sum_{h=1}^{H}\beta_h$. Note that $|S| \leq L$, Theorem 3 can be expressed as $O\left(\max\left\{\sum_{h=1}^{H}\frac{\alpha_h\Delta_{S,\max}}{\Delta_{\mathcal{B}_h,\min}^2}, \frac{(|\mathcal{S}|-\sum_{h=1}^{H}\beta_h)L}{\Delta_{S,\min}}\right\}\log(T)\right)$ where $H$ denotes the times of using the Oracle. In general cases, the size of $\mathcal{R}_h$ cannot be too large, as there are only $M$ base arms in total, and thus a large $\mathcal{R}_h$ is unnecessary. When the size of $\mathcal{G}$ is small, Theorem 3 demonstrates that the algorithm can achieve strong performance.*

## 6 EXPERIMENTS

We compared our algorithm, SortUCB, against several baselines, with results shown in Figure 1. The red line represents CombUCB1 from Kveton et al. (2015), a leading reduction algorithm for the CMAB problem with linear rewards. The green line, labeled MPMAB-s, is based on the MPMAB algorithm Lai & Robbins (1985), applied to $L$ independent MPMAB instances. The blue and orange lines correspond to two versions of SortUCB: the blue line represents Algorithm 1, while the orange line is a variation that uses super arms of size $\lfloor L/2 \rfloor$ for order learning in both the "Elimination" and "Sorting" phases. The plots show cumulative regret as a function of time, averaged over 10 runs, with shaded areas representing empirical standard deviations. Each base arm's reward follows a Bernoulli distribution, $X_{i(\ell),t} \sim \text{Ber}(\mu_{i(\ell)})$.

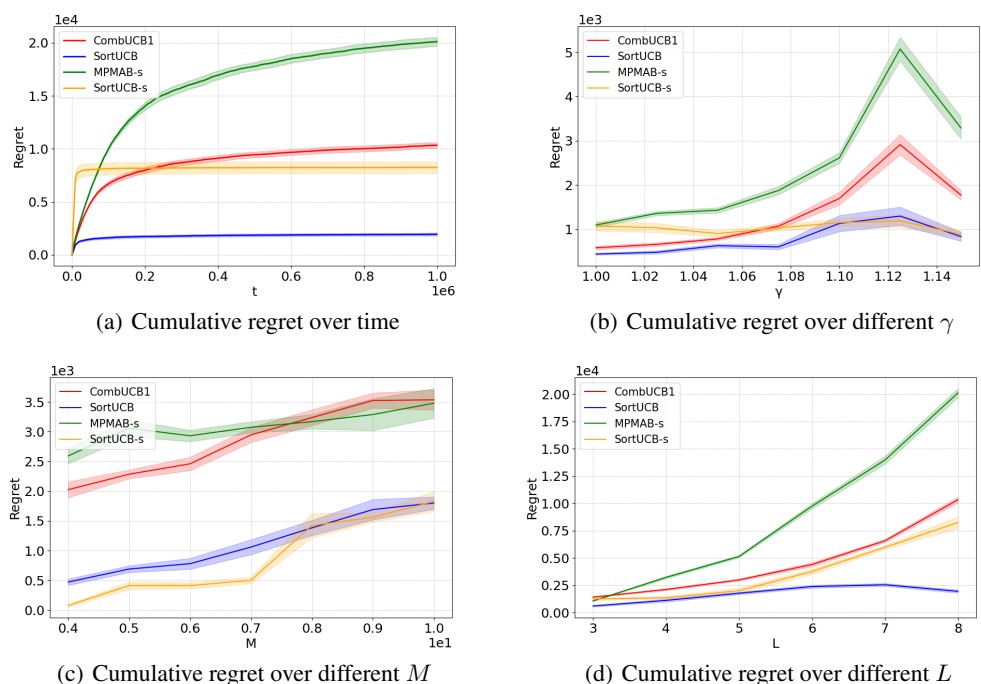

(a) Cumulative regret over time

(b) Cumulative regret over different $\gamma$

(c) Cumulative regret over different $M$

(d) Cumulative regret over different $L$

Figure 1: Experiments

**Experiment 1: Cumulative regret over time.** In this experiment, we compare the regret over time. We set $M = L = 8$ and $T = 10^6$, repeating the experiment for the previously mentioned parameter values. According to Theorem 1, the dominant term of regret for SortUCB stems from the first two phases of where the Algorithm 1 tries to learn the order. Consequently, in the initial time steps in Figure 1(a), our algorithm performs similarly to the baselines. However, as the time

horizon increases, SortUCB rapidly identifies the optimal super arm after learning the order of all base arms, leading to a plateau in regret. In contrast, the regret for CombUCB1 and MPMAB-s grows quickly, as they must still learn a large number of parameters. SortUCB-s performs poorly in this setting because it uses only half of the largest possible number of base arms to learn the order, thus gathering less information than the standard version of SortUCB. This experiment shows that SortUCB outperforms the other three algorithms, particularly as $t$ increases.

**Experiment 2: Cumulative regret over different $\gamma$.** In this experiment, we examine a setting where $\mu_{i(\ell)}/\mu_{L(\ell)} = 1 + \gamma \cdot (L - i)$ for $i \in [M]$ and $\ell \in [L]$, meaning the expected reward for each base arm increases as the super arm size decreases. We consider $M = 6$ and $L = 4$, with $\gamma$ ranging from 0.025 to 0.150. As $\gamma$ increases, the impact of super arm size becomes more significant. Figure 1(b) shows that SortUCB and SortUCB-s outperform CombUCB1 and MPMAB-s, especially at higher $\gamma$ values. While SortUCB-s lags behind SortUCB when $\gamma$ is small due to using fewer base arms, its performance improves as $\gamma$ increases, driven by higher rewards. For all algorithms, cumulative regret decreases significantly at $\gamma = 0.15$, as the larger gap between super arms makes it easier to identify the optimal one. Experiment 2 confirms that our algorithm performs better when the influence of super arm size increases.

**Experiment 3: Fix $T, L$, change $M$.** Here we set $L = 4$ and evaluate multiple instances with varying values of $M$ from 4 to 10. Figure 1(c) shows that the cumulative regret for all algorithms increases at a similar rate, but SortUCB and SortUCB-s consistently outperform the other two baselines. This observation aligns with our theoretical findings, confirming that SortUCB and CMAB algorithms exhibit similar regret growth rates, which are linear with respect to the number of base arms $M$. However, SortUCB and SortUCB-s achieve better performance because they employ more effective policies to learn the structure of the reward distributions.

**Experiment 4: Fix $T, M$, change $L$.** In this experiment, we set $M = 8$ and consider multiple instances with varying values of $L$ from 3 to 8. Figure 1(d) shows that the cumulative regret of SortUCB remains nearly unchanged as $L$ increases and performs significantly better when $L$ is large. This is because the cumulative regret in Algorithm 1 arises from the Elimination Phase and the Sorting Phase, which depends only on $M$ and not $L$. In contrast, the regret of CombUCB1 and MPMAB-s grows rapidly as $L$ increases, since their regret bounds are linear in $ML^2$, which becomes substantially larger as $L$ grows. While SortUCB-s performs better than CombUCB1 and MPMAB-s due to its effective sorting policy, it still lags behind SortUCB because it collects less information per time slot compared to SortUCB. Experiments 3 and 4 demonstrate that our algorithm achieves better performances when dealing with a large number of parameters to learn.

## 7 CONCLUSION

We propose a variant of the classic MAB problem, SSD-CMAB, where the reward of a base arm depends on the size of the super arm it belongs to. Our algorithm, SortUCB, leverages the order preservation property commonly seen in real-world scenarios, and we provide both upper and lower bounds for the SSD-CMAB problem. Experiments show that SortUCB often outperforms traditional CMAB algorithms. Additionally, we extend our model to the SD-CMAB problem, which introduces further complexity. For future work, exploring nonlinear reward functions could expand the applicability of our approach. Furthermore, while we derive a partially tight upper bound, there is room for improvement in both the algorithm and the bounds, particularly in refining the order learning process, which could lead to better performance.

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

## A    LITERATURE REVIEW

Multi-armed bandits, first introduced by Lai & Robbins (1985), have been studied extensively in various generalizations. Among these, the Combinatorial Multi-Armed Bandits (CMAB) model is a key extension of the canonical MAB model and shares some similarities with our proposed framework. CMAB was first introduced by Gai et al. (2012); Cesa-Bianchi & Lugosi (2012), where each action corresponds to pulling a super arm composed of multiple base arms. Subsequently, Chen et al. (2013; 2016b;a) proposed the combinatorial upper confidence bound (CUCB) algorithm for CMAB, achieving near-optimal regret performance (Kveton et al., 2015; Combes et al., 2015a; Merlis & Mannor, 2019). Beyond CUCB, other CMAB algorithms have also been developed (Wang & Chen, 2018; Chen et al., 2014). Moreover, Agarwal et al. (2021) investigated the order-preservation property, which is central to SSD-CMAB.

However, these algorithms are designed for the traditional CMAB problem, where the reward of a base arm follows the same distribution across all super arms. In contrast, SSD-CMAB introduces set-size dependence, where a base arm's outcome depends on the size of the super arm. Specifically, each base arm in SSD-CMAB is associated with $L$ different distributions, one for each possible super arm size, whereas in CMAB, a base arm is tied to a single fixed distribution. While non-linear reward functions in CMAB (Chen et al., 2016b; 2021; Merlis & Mannor, 2019) address cases where super arm rewards are not simple summations of base arm rewards, the base arm outcomes in these models still follow fixed distributions. As such, they cannot capture scenarios where the reward distributions of base arms vary with super arm size.

Our work also relates to bandits with specialized reward structures (Kleinberg & Immorlica, 2018; Hsieh et al., 2022; Basu et al., 2019; Du et al., 2021; Wang et al., 2022). Recharging bandits (Kleinberg & Immorlica, 2018), uplifting bandits (Hsieh et al., 2022), and blocking bandits (Basu et al., 2019) explore how an arm's pulling history affects subsequent reward realizations. A graph-based model is proposed by Wen et al. (2017). Submodular bandits (Yue & Guestrin, 2011; Chen et al., 2017; 2018) are somewhat related to our model since base arm rewards may change based on the presence of other base arms in the super arm, while some of which require the monotonicity property as an assumption (Takemori et al., 2020; Fourati et al., 2024) for the reward function and some do not (Feige et al., 2011; Niazadeh et al., 2021; Fourati et al., 2023). However, one common property that submodular function requires is the submodular property. In contrast, SSD-CMAB only requires the order preservation property, which is different from that in submodular bandits. Hence, our model is able to capture many scenarios that submodular bandits cannot capture.

Finally, bandits with bottleneck rewards (Du et al., 2021) and shareable arms (Wang et al., 2022) study how the super arm structure affects reward realizations, but not how the size of the super arm influences the base arm reward mean itself. To the best of our knowledge, no prior work has explored the Set-Size Dependent reward model proposed in this paper.

## B    REDUCTION VERSION OF CMAB ALGORITHMS

We propose an algorithm that adapts standard CMAB algorithms to effectively address SSD-CMAB problems.

---

**Algorithm 2** Reduction of CMAB Algorithms for SSD-CMAB Problems

---

1: **Input:** Base arm reward means $\mu_{i(\ell)}$ for $i \in [M]$ and $\ell \in [L]$, super arm sets $\mathcal{S}_\ell$ for each size $\ell$.
2: Define a new reward vector $\boldsymbol{\nu}$ of size $D = ML$: $\nu_{i+(\ell-1)M} \leftarrow \mu_{i(\ell)}$ for all $i \in [M]$ and $\ell \in [L]$.
3: Construct transformed super arm sets $\mathcal{S}'_\ell$:

$$\mathcal{S}'_\ell \leftarrow \big\{ \{(\ell-1)M + i_1, (\ell-1)M + i_2, \ldots, (\ell-1)M + i_p\} \mid \{i_1, i_2, \ldots, i_p\} \in \mathcal{S}_\ell \big\}.$$

4: Combine all transformed sets:
$$\mathcal{S}' \leftarrow \bigcup_{\ell \in \{1,2,\ldots,L\}} \mathcal{S}'_\ell.$$

5: Apply any standard CMAB algorithm to the super arm set $\mathcal{S}'$ using the reward vector $\boldsymbol{\nu}$.

---

In the algorithm above, we can treat the problem as a CMAB instance with $D$ base arms and a reward expectation vector $\nu$. The action set for this problem is the transformed set $\mathcal{S}'$, as defined earlier. At each time slot, the learner selects a super arm $S \in \mathcal{S}'$ with $|S| \leq L$. This transformation allows any SSD-CMAB problem to be mapped to a CMAB problem. Since CMAB settings can vary, we focus on the linear reward case, which is highly relevant to our work.

**Theorem 4.** *For a linear reward function in an SSD-CMAB problem (i.e., $r(S) = \sum_{i \in S} \mu_{i(\ell_S)}$), we apply the CombUCB1 algorithm raised by Kveton et al. (2015), which achieves a tight regret bound for linear reward settings, in the second phase of Algorithm 1. The regret bound is given by:*

$$\mathrm{Reg}_T(\mathrm{CombUCB1}, \nu) \leq \sum_{i \in \tilde{E}} K \frac{534}{\Delta_{i,\min}} \log(T) + \left( \frac{\pi^2}{3} + 1 \right) KD,$$

*where*

$$\tilde{E} = [D] - \left\{ \{ (\ell_{S^*} - 1)M + i_1, (\ell_{S^*} - 1)M + i_2, \ldots, (\ell_{S^*} - 1)M + i_p \} \mid i_j \in S^*, j \in [p] \right\}$$

*represents the set of base arms in all suboptimal super arms. Here, $\Delta_{i,\min} = \min_{S \in \mathcal{S}: i \in S, \Delta_S > 0} \Delta_S$ is the smallest gap between the optimal super arm and the best suboptimal super arm containing base arm $i$.*

This formalization highlights the relationship between SSD-CMAB and CMAB problems while leveraging the tight regret guarantees of the CombUCB1 algorithm in CMAB with linear reward function. Notice that Theorem 4 achieves a regret upper bound of $O\left( \frac{ML^2}{\Delta_{S,\min}} \log(T) \right)$ which is worse than our result in 1.

## C  Implementation of Algorithm 1

In this section, we detail the implementation of Algorithm 1 and analyze the computational complexity of its three phases. The overall complexity per time slot is at most $O(M \log(M))$.

**Elimination Phase.**  In the Elimination Phase, the complexity arises from ranking the $L$ smallest $N_{i(L),t}$ base arms and removing base arms, corresponding to lines 4 and 6 in the algorithm. Ranking the base arms can be performed with a complexity of $O(M \log(M))$ per time slot. Deleting base arms requires comparing the lowest upper confidence bound with the largest $L$ upper confidence bounds, as specified in (2), which incurs a complexity of $O(L)$. Therefore, the computational complexity of the Elimination Phase is at most $O(M \log(M))$ per time slot.

**Sorting Phase.**  In the Sorting Phase, the pulled arm in each time slot is fixed to $\mathcal{R}$, and the complexity arises from ranking the $L$ base arms and deleting base arms, similar to the Elimination Phase. Thus, the computational complexity in this phase is $O(L \log(L))$.

**UCB Phase.**  In the UCB Phase, the algorithm selects the super arm with the highest value according to (4) in $\mathcal{A}$. As shown in Appendix D, there are $L$ super arms in $\mathcal{A}$, so the algorithm only needs to identify the largest value among $L$ items, resulting in a computational complexity of $O(L)$.

## D  Proof of Theorem 1

W.L.O.G., according to the order preservation property, we can suppose the reward mean of base arms decreases as the subscript increases (i.e. $\mu_i \succeq \mu_j$ when $i \leq j$). We first give several lemmas.

**Lemma 5.** *Hoeffding's inequality: For independent variables $X_1, X_2, ..., X_n$ with $X_i \in [0,1], i \in [n]$, we have:*

$$\mathbb{P}\left( \left| \frac{1}{n} \sum_{i=1}^{n} (X_i - \mathbb{E}[X_i]) \right| \geq \epsilon \right) \leq 2 \exp\left( -2n\epsilon^2 \right).$$

**Lemma 6.** *Union bound: For a set of $n$ events $A_1, A_2, ..., A_n$, we have:*

$$\mathbb{P}\left( \bigcup_{i=1}^{n} A_i \right) \leq \sum_{i=1}^{n} \mathbb{P}(A_i).$$

**Lemma 7.** *Principle of Inclusion-Exclusion: For event $X_1, X_2, ..., X_n$, we have:*

$$1 - \mathbb{P}\left(\bigcap_{i=1}^n X_i\right) = \mathbb{P}\left(\bigcup_{i=1}^n X_i^c\right).$$

*where $X_i^c$ denotes the complement of event $X_i$.*

**Lemma 8.** *Assume event $G_{i(\ell),t} = \left\{\hat{\mu}_{i(\ell),N_{i(\ell),t}} \in \left[\mu_{i(\ell)} - \sqrt{\frac{2\log(T)}{N_{i(\ell),t}}}, \mu_{i(\ell)} + \sqrt{\frac{2\log(T)}{N_{i(\ell),t}}}\right]\right\}, G = \bigcap_{i\in[M]}\bigcap_{\ell\in[L]}\bigcap_{t\in[T]} G_{i(\ell),t}$. Event $G^c$ denotes the complement part of $G$ (that is, $\mathbb{P}(G) + \mathbb{P}(G^c) = 1$). Then $\mathbb{P}(G^c) \leq \frac{2ML}{T^3}$.*

*Proof of Lemma 8.* We use $G_{i(\ell),t}^c$ to indicate the complement part of $G_{i(\ell),t}$. Using Lemma 6 and 7, probability that event $G^c$ happens is

$$\mathbb{P}(G^c) = 1 - \mathbb{P}(G) = \mathbb{P}\left(\bigcup_{i\in[M]}\bigcup_{\ell\in[L]}\bigcup_{t\in[T]} G_{i(\ell),t}^c\right) \leq \sum_{i\in[M]}\sum_{\ell\in[L]}\sum_{t\in[T]} \mathbb{P}\left(G_{i(\ell),t}^c\right).$$

Firstly we calculate $\mathbb{P}\left(G_{i(\ell),t}^c\right)$ for all $i \in [M], \ell \in [L], t \in [T]$. Using Lemma 6, we can derive

$$\mathbb{P}(G_{i(\ell),t}^c) = \mathbb{P}\left(\hat{\mu}_{i(\ell),N_{i(\ell),t}} < \mu_{i(\ell)} - \sqrt{\frac{2\log(T)}{N_{i(\ell),t}}} \bigcup \hat{\mu}_{i(\ell),N_{i(\ell),t}} > \mu_{i(\ell)} + \sqrt{\frac{2\log(T)}{N_{i(\ell),t}}}\right)$$

$$= \mathbb{P}\left(\left|\hat{\mu}_{i(\ell),N_{i(\ell),t}} - \mu_{i(\ell)}\right| \geq \sqrt{\frac{2\log(T)}{N_{i(\ell),t}}}\right). \tag{9}$$

As $\hat{\mu}_{i(\ell),N_{i(\ell),t}} = \frac{1}{N_{i(\ell),t}}\sum_{s=1}^{N_{i(\ell),t}} X_{i(\ell),s}$ and $\mu_{i(\ell)} = \mathbb{E}\left[\frac{1}{N_{i(\ell),t}}\sum_{s=1}^{N_{i(\ell),t}} X_{i(\ell),s}\right]$, combining Lemma 5 and (9), we obtain

$$\mathbb{P}\left(G_{i(\ell),t}^c\right) \leq \frac{2}{T^4}, \ \mathbb{P}(G^c) = \sum_{i\in[M]}\sum_{\ell\in[L]}\sum_{t\in[T]} \mathbb{P}\left(G_{i(\ell),t}^c\right) \leq \frac{2ML}{T^2}.$$

Here we end the proof of Lemma 8. $\square$

*Proof of Theorem 1.* We use $\text{Reg}_{1,T}(\pi,\nu)$ to denote the regret generated from the Elimination Phase and Sorting Phase in Algorithm 1, and $\text{Reg}_{2,T}(\pi,\nu)$ denotes the regret generated from the UCB Phase. Then

$$\text{Reg}_T(\pi,\nu) = \text{Reg}_{1,T}(\pi,\nu) + \text{Reg}_{2,T}(\pi,\nu).$$

We define $G_{i(\ell),t}$ as below:

$$G_{i(\ell),t} = \left\{\hat{\mu}_{i(\ell),N_{i(\ell),t}} \in \left[\mu_{i(\ell)} - \sqrt{\frac{2\log(T)}{N_{i(\ell),t}}}, \mu_{i(\ell)} + \sqrt{\frac{2\log(T)}{N_{i(\ell),t}}}\right]\right\}, \tag{10}$$

and $G_{i(\ell),t}^c$ denotes the complement part.

As in Algorithm 1, we consider base arms rather than super arms, thus we consider the pulled time slots for each base arm. Since we pull $L$ arms per time slot for sorting, the regret for base arm $i$ each time slot can be seen as $\delta_{i(L)} = \frac{r(S^*)}{L} - \mu_{i(L)}$. Hence

$$\text{Reg}_{1,T}(\pi,\nu) = \sum_{i=1}^M \mathbb{E}\left[N_{i(L),T_1}\right]\delta_{i(L)}$$

holds, where $T_1$ denotes the total time slots before the third cycle. We consider the time slots under $G = \bigcap_{i\in[M]}\bigcap_{\ell\in[L]}\bigcap_{t\in[T]} G_{i(\ell),t}$ and its opposite $G^c$. According to Lemma 8, $\mathbb{P}(G^c) \leq \frac{2ML}{T^3}$ holds, thus the pulling time slots for any base arm $i$ in Elimination Phase is

$$\mathbb{E}\left[N_{i(L),T_1}\right] \leq \mathbb{E}\left[N_{i(L),T_1}^G\right] + \frac{2ML}{T^2}.$$

Then we can just consider the time slots for $\mathbb{E}\left[N^G_{i(L),T_1}\right]$. W.L.O.G, we assume $\mu_{1(L)} \geq \mu_{2(L)} \geq ... \geq \mu_{M(L)}$. Below we consider cases when $L < M$ holds, as for the case that $L = M$, the first cycle in Algorithm 1 does not run, and can be easily deduced from the proof below. Afterwards we give a lemma ensuring base arms we eliminate cannot be concluded in the optimal super arm with high probability.

**Lemma 9.** *If event $G$ happens, the base arms we eliminate in line 7 cannot be concluded in $S^*$.*

*Proof of Lemma 9.* Combining (10) and the condition (2), we have

$$\mu_{i(L)} \leq \hat{\mu}_{i(L),N^G_{i(L),T_1}} + \sqrt{\frac{2\log(T)}{N^G_{i(L),T_1}}} < \hat{\mu}_{j(L),N^G_{i(L),T_1}} - \sqrt{\frac{2\log(T)}{N^G_{i(L),T_1}}} \leq \mu_{j(L)} \qquad (11)$$

hold for at least $L$ different base arms $j$. Therefore, arms that we eliminate cannot be any of the first $L$ of base arms, meaning they cannot be concluded in the optimal super arm. That ends the proof. $\qquad\square$

According to Lemma 9, in Elimination Phase, we successfully find the first $L$ arms. Therefore, we can always eliminate the suboptimal base arms (denoted by $[M] - [L] = \{L+1, L+2, ..., M\}$). We consider the time slots that the first $L$ base arms are pulled as well as the other base arms separately.

For base arm in $[M] - [L]$, as they will eventually be eliminated in this cycle, we can bound their pulled time slots. Consider base arm $i \in [M] - [L]$, if it is not eliminated, as the opposite of (11),

$$\hat{\mu}_{i(L),N^G_{i(L),T_1}} + \sqrt{\frac{2\log(T)}{N^G_{i(L),T_1}}} \geq \hat{\mu}_{j(L),N^G_{j(L),T_1}} - \sqrt{\frac{2\log(T)}{N^G_{j(L),T_1}}}$$

hold for at least $M - L + 1$ base arms $j$ in $[M]$. We use $E$ to denote the set for all possible base arms satisfying (11), where $|E| \geq M - L + 1$. According to (10), this means

$$\mu_{i(L)} + 2\sqrt{\frac{2\log(T)}{N^G_{i(L),T_1}}} \geq \mu_{j(L)} - 2\sqrt{\frac{2\log(T)}{N^G_{j(L),T_1}}} \qquad (12)$$

holds for base arms in $E$. As for any two base arms $i$ and $j$, their pulling time slots differs no more than 1 according to the uniform pulling, which lead to

$$\max(N^G_{i(L),T_1}, N^G_{j(L),T_1}) \leq \frac{32\log(T)}{\Delta^2_{i(L),j(L)}} + 1.$$

Since there are $|E|$ choices for arm $j$, we have $N^G_{i(L),T_1} \leq \min_{j \in E} \frac{32\log(T)}{\Delta^2_{i(L),j(L)}} + 1$. As $i \in [M] - [L] + 1$ and $|E| \geq M - L + 1$, at least 1 base arm in $[L]$ that is in $E$. Thus, it holds that $\max_{j \in B} \Delta^2_{i(L),j(L)} \geq \Delta^2_{i(L),L(L)}$. Therefore,

$$N^G_{i(L),T_1} \leq \frac{32\log(T)}{\Delta^2_{i(L),L(L)}} + 1$$

holds for all base arm $i \in [M] - [L]$. Therefore when $M = L$, $[M] - [L] = \phi$, meaning Elimination Phase does not run in this case.

For base arm $i \in [L]$, as we cannot eliminate them under $G$, and once all base arms in $[M] - [L]$ have been eliminated, the first cycle ends. As we have declared before, pulled time slots for two base arms not eliminated do not differ than 1, we have

$$N^G_{i(L),t} \leq \max_{j \in [M]-[L]} \frac{32\log(T)}{\Delta^2_{j(L),L(L)}} + 2 = \frac{32\log(T)}{\Delta^2_{(L+1)(L),L(L)}} + 2,$$

where $\Delta_{(M+1)(L),M(L)} = \infty$ when $L = M$.

In Sorting Phase, as we have found the first $L$ arms, and our goal is to sort for these $L$ base arms. We give Lemma 10 to ensure we can get the right sequence of the first $L$ base arms.

**Lemma 10.** *If event $G$ happens, the second cycle in Algorithm 1 can get the right order of the first $L$ base arms with high probability.*

*Proof of Lemma 10.* According to (10) and the condition (2), we have $\mu_{i(L)} < \mu_{j(L)}$ for all base arms $j \in B - i$ holds. That means base arm $j$ is the worst base arm in $B$, thus we can get the right order for the first $L$ base arms. $\square$

With Lemma 10, we can continue our proof. Consider base arm $i$ which is still in $E'$, it means

$$\mu_{i(L)} + 2\sqrt{\frac{2\log(T)}{N^G_{i(L),T_1}}} \geq \mu_{j(L)} - 2\sqrt{\frac{2\log(T)}{N^G_{j(L),T_1}}}$$

hold for all $j \neq i$ and $j \in E'$.

Since we uniformly pull all base arms, $N_{i(K)}$ do not differ more than 1 between any two base arms in the first $L$ arms. Thus,

$$N^G_{i(L),T_1} \leq \frac{32\log(T)}{\Delta^2_{i(L),j(L)}} + 1 \leq \frac{32\log(T)}{\min_{j \in [L-1]}\left(\Delta^2_{j(L),(j+1)(L)}\right)} + 1. \tag{13}$$

Therefore, $N^G_{i(L),T_1} \leq \frac{32\log(T)}{\min_{j \in [L-1]}\left(\Delta^2_{j(L),(j+1)(L)}\right)} + 2$ holds for all $i \in [L]$. Since we have proved that $N^G_{i(L),T_1} \leq \frac{32\log(T)}{\Delta^2_{(L+1)(L),L(L)}} + 2$ in the cycle, we have

$$N^G_{i(L),T_1} \leq \frac{32\log(T)}{\min_{j \in [L]}\left(\Delta^2_{j(L),(j+1)(L)}\right)} + 2$$

for all $i \in [L]$. Combining with the definition of $N_{i(L),T_1}$, we have

$$N_{i(L),T_1} \leq \begin{cases} \frac{32\log(T)}{\min_{j \in [L]}\left(\Delta^2_{j(L),(j+1)(L)}\right)} + 2 + \frac{2ML}{T^2}, & \text{if } i \in [L], \\ \frac{32\log(T)}{\Delta^2_{i(L),L(L)}} + 1 + \frac{2ML}{T^2}, & \text{if } i \in [M] - [L]. \end{cases}$$

As a result,

$$\text{Reg}_{1,T}(\pi,\nu) = \sum_{i=1}^{M} N_{i(L),T_1}\delta_{i(L)} \leq \sum_{i \in [M]-[L]} \frac{32\log(T)}{\Delta^2_{i(L),L(L)}}\delta_i + \frac{32\log(T)}{\min_{j \in [L]}\left(\Delta^2_{j(L),(j+1)(L)}\right)}\Delta_{\{1,2,\dots,L\}}$$

$$+ \left(2 + \frac{2ML}{T^2}\right)\sum_{i=1}^{M}\delta_i. \tag{14}$$

That is the end of proof of $\text{Reg}_{1,T}(\pi,\nu)$.

Below we prove the bound for $\text{Reg}_{2,T}(\pi,\nu)$. In UCB Phase, our intuition is seeing each super arm as a single item. We use $\mu_S = \sum_{i \in S}\mu_{i(\ell_S)}$ to denote the reward expectation for super arm $S$ and $\hat{\mu}_{S,N_{S,t}}$ to denote the unbiased estimate for super arm $S$ in the first $t$ time slots, while $N_{S,t}$ indicates the chosen times for super arm $S$ as a whole since the start of the second cycle, which is initialized to zero in line 11 in Algorithm 1. It is simple to show that

$$\text{Reg}_{2,T}(\pi,\nu) = \sum_{S \in \mathcal{A}} \mathbb{E}\left[N_{S,T}\right]\Delta_S. \tag{15}$$

First we give a lemma that ensures the optimal super arm can be in $\mathcal{A}$ with high probability.

**Lemma 11.** *If event $G$ happens, the optimal super arm (denoted by $S^*$) must be concluded in $\mathcal{A}$.*

*Proof of Lemma 11.* As we defined, event $G$ means $\mu_{i(\ell)} - \sqrt{\frac{2\log(T)}{N_{i(\ell),t}}} \leq \hat{\mu}_{i(\ell),N_{i(\ell),t}} \leq \mu_{i(\ell)} + \sqrt{\frac{2\log(T)}{N_{i(\ell),t}}}$ holds for each $i \in [M], \ell \in [L], t \in [T]$. For each super arm size $\ell \in [L]$, as we have learnt the best $\ell$ base arms and combine them as a super arm in $\mathcal{A}$, Lemma 11 holds obviously. $\square$

Now we continue our proof of bound in $\text{Reg}_{2,T}(\pi, \nu)$. As we have proved in previous,

$$\text{Reg}_{2,T}(\pi, \nu) = \sum_{S \in \mathcal{A}} \mathbb{E}\left[N_{S,T}\right] \Delta_S \leq \sum_{S \in \mathcal{A}} \mathbb{E}\left[N_{S,T}^G\right] \Delta_S + \frac{2ML}{T^2} \sum_{S \in \mathcal{A}} \Delta_S$$

$$= \sum_{\substack{S \in \mathcal{A} \\ S \neq S^*}} \mathbb{E}\left[N_{S,T}^G\right] \Delta_S + \frac{2ML}{T^2} \sum_{S \in \mathcal{A}} \Delta_S. \tag{16}$$

The last equation is because of $\Delta_{S^*} = 0$. Thus, we only need to prove the upper bound for $\mathbb{E}\left[N_{S,T}^G\right]$. We first define another event $\tilde{G}_S$ which we need in our proof:

$$\tilde{G}_S = \left\{ \mu_{S^*} < \min_{t \in [T]} \hat{\mu}_{S^*, N_{S^*,t}} + \sqrt{\frac{2|S^*|\log(T)}{N_{S^*,t}}} \right\} \cap \left\{ \hat{\mu}_{S,u_S} + \sqrt{\frac{2|S|\log(T)}{u_S}} < \mu_{S^*} \right\},$$

where $u_S \in [T]$ is a constant to be chosen later. Two lemmas are introduced for our derivation,

**Lemma 12.** *If $\tilde{G}_S$ occurs, then $N_{S,T} \leq u_S$.*

**Lemma 13.** *$\tilde{G}_S^c$, meaning the complement part of $\tilde{G}_S$, happens with low probability.*

As $N_{S,T}^G \leq T$, we use $N_{S,t}^{G \cap \tilde{G}_S}$ to denote the pulling times for super arm $S$ in the first $t$ time slots in the second cycle with event $G$ and $\tilde{G}_S$ both happening, while $N_{S,t}^{G \cap \tilde{G}_S^c}$ means that only event $G$ happens while event $\tilde{G}_S$ does not happen. Then,

$$\mathbb{E}\left[N_{S,T}^G\right] = \mathbb{E}\left[\mathbb{I}(\tilde{G}_S) N_{S,T}^{G \cap \tilde{G}_S}\right] + \mathbb{E}\left[\mathbb{I}(\tilde{G}_S^c) N_{S,T}^{G \cap \tilde{G}_S^c}\right] \leq \mathbb{E}\left[N_{S,T}^{G \cap \tilde{G}_S}\right] + T \cdot \mathbb{P}\left(\tilde{G}_S^c\right). \tag{17}$$

*Proof of Lemma 12.* Assuming that $\tilde{G}_S$ occurs with $N_{S,T} \geq u_S$. That means there exists $t \in T$ s.t. $N_{S,t-1} = u_S$ while $A_t = S$, where $A_t$ means the chosen super arm at time slot $t$. Hence we have:

$$\hat{\mu}_{S,N_{S,t-1}} + \sqrt{\frac{2|S|\log(T)}{N_{S,t-1}}} \leq \mu_{S^*} \leq \hat{\mu}_{S^*,N_{S,t-1}} + \sqrt{\frac{2|S^*|\log(T)}{N_{S,t-1}}}. \tag{18}$$

That means in time slot $t$ we should choose super arm $S^*$ rather than arm $S$, which is a contradiction. $\square$

*Proof of Lemma 13.* The complement part of $\tilde{G}_S$ is

$$\tilde{G}_S^c = \left\{ \mu_{S^*} \geq \min_{t \in [T]} \left( \hat{\mu}_{S^*,N_{S^*,t}} + \sqrt{\frac{2|S^*|\log(T)}{N_{S^*,t}}} \right) \right\} \cup \left\{ \hat{\mu}_{S,u_S} + \sqrt{\frac{2|S|\log(T)}{u_S}} \geq \mu_{S^*} \right\}. \tag{19}$$

We begin with bounding the first part of $\tilde{G}_S^c$. As

$$\left\{ \mu_{S^*} \geq \min_{t \in [T]} \left( \hat{\mu}_{S^*,t} + \sqrt{\frac{2|S^*|\log(T)}{N_{S^*,t}}} \right) \right\} \subset \left\{ \mu_{S^*} \geq \min_{t \in [T]} \left( \hat{\mu}_{S^*,t} + \sqrt{\frac{2|S^*|\log(T)}{t}} \right) \right\}$$

$$= \bigcup_{t \in [T]} \left\{ \mu_{S^*} \geq \hat{\mu}_{S^*,t} + \sqrt{\frac{2|S^*|\log(T)}{t}} \right\}. \tag{20}$$

Thus, using Lemma 6, we have

$$
\mathbb{P}\left(\mu_{S^*} \geq \min_{t \in [T]}\left(\hat{\mu}_{S^*,t} + \sqrt{\frac{2|S^*|\log(T)}{N_{S^*,t}}}\right)\right) \leq \mathbb{P}\left(\bigcup_{t \in [T]}\left\{\mu_{S^*} \geq \hat{\mu}_{S^*,t} + \sqrt{\frac{2|S^*|\log(T)}{t}}\right\}\right)
$$

$$
\leq \sum_{t=1}^{T} \mathbb{P}\left(\mu_{S^*} \geq \hat{\mu}_{S^*,t} + \sqrt{\frac{2\,|S^*|\log(T)}{t}}\right) \leq \frac{1}{T^3}, \tag{21}
$$

which is a low probability if $T$ is chosen large enough. For the last inequality in (21), we use Lemma 5 with $t|S^*|$ independent samples.

Next we bound the second part of $\tilde{G}_S^c$. As $u_S$ is a parameter undetermined, we assume it is large enough that $\Delta_S - \sqrt{\frac{2|S|\log(T)}{u_S}} \geq c\Delta_S$, where $c \in (0,1)$ will be chosen later. Thus,

$$
\mathbb{P}\left(\hat{\mu}_{S,u_S} + \sqrt{\frac{2|S|\log(T)}{u_S}} \geq \mu_{S^*}\right) = \mathbb{P}\left(\hat{\mu}_{S,u_S} - \mu_S \geq \Delta_S - \sqrt{\frac{2|S|\log(T)}{u_S}}\right)
$$

$$
\leq \mathbb{P}(\hat{\mu}_{S,u_S} - \mu_S \geq c\Delta_S) \leq \exp\left(-2c^2\Delta_S^2\frac{u_S}{|S|}\right). \tag{22}
$$

Taking together (21) and (22),

$$
\mathbb{P}\left(\tilde{G}_S^c\right) \leq \frac{1}{T^3} + \exp\left(-2c^2\Delta_S^2\frac{u_S}{|S|}\right).
$$

Here we end the proof of Lemma 13. □

As we have proved in (17)

$$
\mathbb{E}\left[N_{S,T}^G\right] \leq u_S + T\left(\frac{1}{T^3} + \exp\left(-2c^2\Delta_S^2\frac{u_S}{|S|}\right)\right) = u_S + T\exp\left(-2c^2\Delta_S^2\frac{u_S}{|S|}\right) + \frac{1}{T^2}.
$$

Choosing $u_S = \lceil\frac{2|S|\log(T)}{(1-c)^2\Delta_S^2}\rceil$ and $c = 1/2$, then

$$
\mathbb{E}\left[N_{S,T}\right] \leq 3 + \frac{8|S|\log(T)}{\Delta_S^2}.
$$

Considering (16), we can give an upper bound for $\mathrm{Reg}_{2,T}(\pi,\nu)$,

$$
\mathrm{Reg}_{2,T}(\pi,\nu) \leq \sum_{S \in \mathcal{A}, S \neq S^*}\left(3\Delta_S + \frac{8|S|\log(T)}{\Delta_S}\right) + \frac{ML}{T^2}\sum_{S \in \mathcal{A}}\Delta_S. \tag{23}
$$

Combining with (14) and (23), we have $\mathrm{Reg}_T(\pi,\nu) \leq$

$$
\sum_{i \in [M]-[L]}\frac{32\log(T)}{\Delta_{i(L),L(L)}^2}\delta_{i(L)} + \frac{32\log(T)}{\min_{j \in [L]}\left(\Delta_{j(L),(j+1)(L)}^2\right)}\Delta_{\{1,2,...,L\}} + \sum_{S \in \mathcal{A}-S^*}\frac{8|S|\log(T)}{\Delta_S}
$$

$$
+ \left(\frac{2ML}{T^2} + 2\right)\sum_{i=1}^{M}\delta_{i(L)} + \left(\frac{2ML}{T^2} + 3\right)\sum_{S \in \mathcal{A}}\Delta_S,
$$

where $\Delta_{(M)(L),(M+1)(L)} = \infty$ when $L = M$. Simplify this bound and we then end of proof of Theorem 1. □

## E  PROOF OF THEOREM 2

We first give several lemmas that we need in the proof.

**Lemma 14.** *Divergence decomposition: Let $\nu = (P_1, \ldots, P_m)$ denotes the reward distributions for an $m$-armed bandit problem and $\nu' = (P'_1, \ldots, P'_m)$ denotes another. For a fixed policy $\pi$, we have:*

$$D(\mathbb{P}_\nu, \mathbb{P}_{\nu'}) = \sum_{i=1}^m \mathbb{E}_\nu[T_i(T)]D(P_i, P'_i).$$

*where $\mathbb{P}_\nu = \mathbb{P}_{\nu\pi}$ and $\mathbb{P}_{\nu'} = \mathbb{P}_{\nu'\pi}$ be the probability measures on the canonical bandit model induced by the $T$-time slot interconnection of $\pi$ and $\nu$ (or $\nu'$).*

**Lemma 15.** *Bretagnolle-Huber inequality: Let $P$ and $Q$ be probability measures on the same measure space $(\Omega, \mathcal{F})$, and let $A \in \mathcal{F}$ be an arbitrary event. Then,*

$$P(A) + Q(A^c) \geq \frac{1}{2} \exp\left(-D(P, Q)\right),$$

*where $A^c = \Omega - A$ denotes the complement part of $A$.*

**Lemma 16.** *Let $\mathcal{E} = \mathcal{M}_1 * \cdots * \mathcal{M}_m$ and $\pi \in \Pi_{cons}(\mathcal{E})$ be a consistent policy over $\mathcal{E}$. Then, for all $\nu = (P_i)_{i=1}^m \in \mathcal{E}$, it holds that*

$$\lim_{T \to \infty} \inf \frac{R_T(\pi, \nu)}{\log(T)} \geq c^*(\nu, \mathcal{E}) = \sum_{i:\Delta_i > 0} \frac{\Delta_i}{d_{\inf}(P_i, \mu^*, \mathcal{M}_i)},$$

*where $\Delta_i$ is the suboptimality gap of the $i$-th arm in $\nu$ and $\mu^*$ denotes the mean reward of the optimal arm.*

The proof of above lemmas can be found in Lattimore & Szepesvári (2020). Below we give proof about Lemma 3 and 4.

*Proof of Lemma 3.* First, we rearrange term $\nu = (P_{1(1)}, \ldots, P_{M(1)}, \ldots \ldots, P_{1(L)}, \ldots, P_{M(L)})$ to denote the SSD-CMAB instance, just for convenience. Since the optimal super arm has $\ell^*$ base arms, we call the first $\ell^*$ base arms "optimal base arms" and the rest "suboptimal base arms". Consider a suboptimal base arm $i$ (i.e. $i > \ell^*$), let $\varepsilon > 0$. We define $\nu' = (P'_{j(\ell)})_{\substack{j \in [M] \\ \ell \in [L]}}$ satisfying that for each $\ell \in [L]$, $P'_{j(\ell)} = P_{j(\ell)}$ when $j \neq i$, $P_{i(\ell)}$ is the $i$-th term in some vector $\mathcal{P} \in \mathcal{M}_\ell$ such that $D(P_{i(\ell)}, P'_{i(\ell)}) \leq d_{\inf}(P_{i(\ell)}, \mu_{\ell^*(\ell)}, \mathcal{M}_\ell) + \varepsilon$ and $\mu(P_{\ell^*(\ell)}) < \mu(P'_{i(\ell)}) < \mu(P_{(\ell^*-1)(\ell)})$. Let $\mu' \in R^{ML}$ be the vector of means of distributions of $\nu'$. By Lemma 14, we have $D(\mathbb{P}_{\nu\pi}, \mathbb{P}_{\nu'\pi}) \leq \sum_{\ell=1}^L \mathbb{E}_{\nu\pi}[T_{i(\ell)}(T)](d_{\inf}(P_{i(\ell)}, \mu_{\ell^*(\ell)}, \mathcal{M}_\ell) + \varepsilon)$.

By Lemma 15, for any event $A$,

$$\mathbb{P}_{\nu\pi}(A) + \mathbb{P}_{\nu'\pi}(A^c) \geq \frac{1}{2} \exp\left(-\sum_{\ell=1}^L \mathbb{E}_{\nu\pi}\left[T_{i(\ell)}(T)\right]\left(d_{\inf}(P_{i(\ell)}, \mu_{\ell^*(\ell)}, \mathcal{M}_\ell) + \varepsilon\right)\right). \quad (24)$$

Suppose $\ell^{*'}$ is the size of optimal super arm in $\nu'$. Obviously $\ell^{*'} \geq \ell^*$, since the means for the best super arm in class $\ell$ where $\ell < \ell^*$ do not change and are worse than super arm $\{1, \ldots, \ell^*\}$. We assume the optimal super arm in $\nu'$ is unique (which can be implemented by fine tuning $\nu'$). Choose $A = \{T_{i(\ell^{*'})}(T) \geq \frac{T}{2}\}$. Let $R_T = R_T(\pi, \nu), R'_T = R_T(\pi, \nu')$.

For $\nu$, each time a super arm in class $\ell^{*'}$ concluding base arm $i$ is pulled, regret increases by at least $\Delta_1 = \min\{\Delta_{i(\ell^*), \ell^*(\ell^*)}, \min_{\ell \neq \ell^*} \Delta_{\{1,\ldots,\ell\}}\}$ (where the first term comes from cases when $\ell^{*'} = \ell^*$, and the second term comes from cases when $\ell^{*'} \neq \ell^*$). Then $R_T \geq \frac{T}{2} \cdot P(A) \cdot \Delta_1$.

For $\nu'$, base arm $i$ is in the optimal super arm in $\ell^{*'}$ since $\mu\left(P'_{i(\ell^{*'})}\right) > \mu\left(P_{\ell^*(\ell^{*'})}\right) \geq \mu\left(P_{\ell^{*'}(\ell^{*'})}\right)$. Each time a super arm in class $\ell^{*'}$ not concluding base arm $i$ is pulled, the regret increases by at least $\Delta_2 = \min\{\mu\left(P'_{i(\ell^{*'})}\right) - \mu\left(P_{\ell^{*'}(\ell^*)}\right), \min_{\ell \neq \ell^{*'}} \Delta'_\ell\}$, where $\Delta'_\ell$ denotes the

gap for the optimal super arm in class $\ell$ in $\nu'$ which is greater than $0$ since the optimal super arm in $\nu'$ is unique. Therefore, $R'_T \geq \frac{T}{2} \cdot P(A^c) \cdot \Delta_2$.

Combining $R_T$, $R'_T$ and inequality (24), we have

$$R_T + R'_T \geq \frac{T}{2}\left(P(A) + P(A^c)\right)\min(\Delta_1, \Delta_2)$$

$$\geq \frac{T}{4}\min(\Delta_1, \Delta_2)\exp\left(-\sum_{\ell=1}^{L}\mathbb{E}_{\nu\pi}\left[T_{i(\ell)}(T)\right]\left(d_{\inf}(P_{i(\ell)}, \mu_{\ell^*(\ell)}, \mathcal{M}_\ell) + \varepsilon\right)\right). \qquad (25)$$

Rearranging (25) and combining Definition 1, we have:

$$\lim_{T\to\infty}\frac{\sum_{\ell=1}^{L}\mathbb{E}_{\nu\pi}[T_{i(\ell)}(T)](d_{\inf}(P_{i(\ell)}, \mu_{\ell^*(\ell)}, \mathcal{M}_\ell) + \varepsilon)}{\log(T)} \geq 1. \qquad (26)$$

Below we consider how the regret are composed. As $R_T \geq \sum_{i=\ell^*+1}^{M} R_{T,i}$, where $R_{T,i}$ denotes the regret from pulling a suboptimal base arm $i$. Consider base arm $i$ can be pulled in super arms from different class, we need to give a lower bound for each $R_{T,i}$. For a suboptimal base arm $i$ pulled in super arm from class $\ell$, as the optimal super arm in class $\ell$ is $\{1, \ldots, \ell\}$ and the regret for choosing which is $\Delta_{\{1,\ldots,\ell\}}$, then the regret for choosing super arm concluding base arm $i$ is at least $\frac{\Delta_{\{1,\ldots,\ell\}}}{\ell} + \max(\mu_{\ell(\ell)} - \mu_{i(\ell)}, 0)$ (the first term comes from the regret generated by the optimal super arm in class $\ell$ divided into $\ell$ parts, and the second term comes from the regret generated by choosing base arm $i$).

Sum them up, we have

$$R_{T,i} \geq \sum_{\ell=1}^{L}\mathbb{E}\left[T_{i(\ell)}(T)\right]\left(\frac{\Delta_{\{1,\ldots,\ell\}}}{\ell} + \max\left(\mu_{\ell(\ell)} - \mu_{i(\ell)}, 0\right)\right)$$

$$\geq \min_{\ell\in[L]}\left(\frac{\Delta_{\{1,\ldots,\ell\}}/\ell + \max\left(\mu_{\ell(\ell)} - \mu_{i(\ell)}, 0\right)}{d_{\inf}(P_{i(\ell)}, \mu_{\ell^*(\ell)}, \mathcal{M}_\ell) + \varepsilon}\right) \cdot \sum_{\ell=1}^{L}\mathbb{E}\left[T_{i(\ell)}(T)\right]\left(d_{\inf}\left(P_{i(\ell)}, \mu_{\ell^*(\ell)}, \mathcal{M}_\ell\right) + \varepsilon\right).$$

Combine inequality (26), for all $i \geq \ell^* + 1$, it holds that

$$\lim_{T\to\infty}\frac{R_{T,i}}{\log(T)} \geq \min_{\ell\in[L]}\left(\frac{\Delta_{\{1,\ldots,\ell\}}/\ell + \max\left(\mu_{\ell(\ell)} - \mu_{i(\ell)}, 0\right)}{d_{\inf}\left(P_{i(\ell)}, \mu_{\ell^*(\ell)}, \mathcal{M}_\ell\right) + \varepsilon}\right).$$

Thus, sum up for all $\ell \geq \ell^* + 1$, we have

$$\lim_{T\to\infty}\frac{R_T}{\log(T)} \geq \sum_{i=\ell^*+1}^{M}\min_{\ell\in[L]}\left(\frac{\Delta_{\{1,\ldots,\ell\}}/\ell + \max\left(\mu_{\ell(\ell)} - \mu_{i(\ell)}, 0\right)}{d_{\inf}(P_{i(\ell)}, \mu_{\ell^*(\ell)}, \mathcal{M}_\ell) + \varepsilon}\right).$$

We end the proof when $\varepsilon$ tends to zero. $\qquad\square$

*Proof of Lemma 4.* First, we consider a map $v : \Pi_{cons}(\mathcal{E}) \to \Pi'_{cons}(\mathcal{E})$ ($\Pi'_{cons}(\mathcal{E})$ will be defined later) that maps a policy $\pi$ to $v(\pi)$ as the following way: In time slot $t$, assume $\pi$ chooses the super arm in class $\ell$, policy $v(\pi)$ chooses the best super arm in class $\ell$ (i.e. super arm $\{1, \ldots, \ell\}$ if the mean reward of base arm decreases in subscript order). The set $\Pi'_{cons}(\mathcal{E})$ concludes all consistent policies which only chooses super arms in $\{\{1, \ldots, \ell\}|\ell \in [L]\}$, over $\mathcal{E}$.

Obviously, policy $v(\pi)$ is always choosing a better super arm than $\pi$. Thus, $R_T(\pi, \nu) \geq R_T(v(\pi), \nu)$ holds. Therefore, we only need to prove for all $\nu = (\mathcal{P}_\ell)_{\ell=1}^{L} \in \mathcal{E}$ and $\pi \in \Pi'_{cons}(\mathcal{E})$, Lemma 4 holds.

Consider $Q_\ell = \sum_{j=1}^{\ell} P_{j(\ell)}$ which denotes the sum of distributions for the best $\ell$ base arms in class $\ell$. Using Lemma 3 on the $L$ distributions $Q_1, \ldots, Q_L$, Lemma 4 can be verified on $\pi \in \Pi_{cons}(\mathcal{E})'$. Therefore, as $R_T(\pi, \nu) \geq R_T(v(\pi), \nu)$ holds, and Lemma 4 holds. $\qquad\square$

| Class ($k$) | Super arm set ($\mathcal{S}_k$) |
|---|---|
| $k = 1$ | $S_1 = \{1, 2\}, S_2 = \{2, 3\}, S_3 = \{1, 3\}, S_4 = \{4\}$ |
| $k = 2$ | $S_5 = \{1, 2, 3\}, S_6 = \{2, 3, 4\}, S_7 = \{5\}$ |
| $k = 3$ | $S_8 = \{1, 5\}, S_9 = \{4, 5\}, S_{10} = \{3, 5\}, S_{11} = \{2\}$ |
| $k = 4$ | $S_{12} = \{1, 2, 3, 4\}, S_{13} = \{1, 2, 3, 5\}, S_{14} = \{1, 4, 5\}, S_{15} = \{2, 3, 4, 5\}$ |

Table 1: An instance of SD-CMAB problem

# F  SET DEPENDENT COMBINATORIAL MULTI-ARMED BANDIT

## F.1  DETAILED SETTING OF SD-CMAB

An instance of a SD-CMAB problem involves $M$ *base arms*. We consider a time horizon of length $T$. Let $\mathcal{S}$ denote the restricted action set $\mathcal{S} \subseteq \{S \subseteq [M] : |S| \leq L\}$ where $L$ denotes the maximum number of base arms in a super arm. In each time slot, the learner plays a super arm $S \in \mathcal{S}$, which is a set of base arms.

In SD-CMAB, the super arm $S \in \mathcal{S}$ affects the distributions for base arms in $S$. As all super arms are in the set $\mathcal{S}$, we divide $\mathcal{S}$ into $K$ different classes $\{\mathcal{S}_1, ..., \mathcal{S}_K\}$. That is, $\bigcup_{i \in [K]} \mathcal{S}_i = \mathcal{S}$, $\mathcal{S}_k \cap \mathcal{S}_{k'} = \emptyset$ for two different $k, k' \in [L]$, where $\emptyset$ denotes an empty set. For any base arm $i$, when it is pulled in super arm $S \in \mathcal{S}_k$, it obeys the distribution related to class $k$, denoted by $\mathbb{P}_{i(k)}$. Notation $k_S$ indicates the class that super arm $S$ is in. Without loss of generality, we assume the rewards of the base arms are $[0, 1]$-valued. We use $\mu_{i(k)}$ to denote the reward expectation for arm $i$ in distribution $\mathbb{P}_{i(k)}$ in class $k$ while $\boldsymbol{\mu}_i = (\mu_{i(1)}, \mu_{i(2)}, ..., \mu_{i(K)})$ indicates the reward vector for base arm $i$ in all classes. When pulled in a super arm from a different class, base arm $i$ obeys different distributions, corresponding to the "Set Dependent".

We use $N_{i(k),t}$ to indicate the number of times that base arm $i$ has been pulled with super arms in class $k$ until time slot $t$. $R(S_t)$ denotes the reward of the chosen super arm at $t$-th time slot, where $r(S_t) = \mathbb{E}[R(S_t)]$ shows its expectation. We consider the linear reward function in which the reward function is $R_(S_t) = \sum_{i \in S} X_{i(k_S), N_{i(k_S),t}}$, where variable $X_{i(k),t}$ indicates the outcome of base arm $i$ in its $t$-th trial with distribution $\mathbb{P}_{i(k)}$. Thus, $\mathbb{E}[X_{i(k_S), N_{i(k_S),t}}] = \mu_{i(k_S)}$. We consider the semi-bandit feedback, which means the learner can observe the reward for any base arm in the super arm it pulls.

As mentioned in the introduction, the order preservation property also exists in SSD-CMAB:
**Order preservation.** For any class $k \in [K]$, the order of reward expectations is fixed across different base arms. That is, $\mu_{i(k)} \leq \mu_{j(k)}$ if and only if $\mu_{i(k')} \leq \mu_{j(k')}$, where $i, j \in [M]$, $k, k' \in [K]$.

Table 1 gives an instance of general SD-CMAB framework with 5 base arms ($M = 5$). The size of $\mathcal{S}$ is 15 and the number of classes is 4 ($K = 4$), meaning these 15 super arms are separated into 4 different classes $\mathcal{S}_1$ to $\mathcal{S}_4$. Fix a base arm, its reward expectation keeps unchanged when it is pulled in super arms from a same class, while that may change when it is pulled in super arms from different classes. For example, consider base arm 3, its reward expectation is same in super arm $S_2$ and $S_3$, but can be different in super arm $S_{10}$ or $S_{13}$.

The objective is to find an algorithm $\pi$ to minimize $\text{Reg}_T(\pi, \nu)$ on SD-CMAB instance $\nu$ which is defined as

$$\text{Reg}_T(\pi, \nu) = T \cdot r(S^*) - \mathbb{E}\left[\sum_{t=1}^{T} r(S_t)\right], \qquad (27)$$

where $S^*$ denotes the optimal super arm in $\mathcal{S}$.

## F.2  DETAILED EXPLANATION FOR SORTUCB-SD

In this appendix we introduce the detailed explanation for SortUCB-SD. We first propose the $(n_1, n_2)$-efficiency Oracle designed for our algorithm.

| Ordered base arms | Explored super arms | Number of eliminated super arms |
|:---:|:---:|:---:|
| $\mathcal{B} = \{1,2,3\}$ | $S_5$ | 3 |
| $\mathcal{B} = \{1,2,3,4\}$ | $S_{12}$ | 6 |
| $\mathcal{B} = \{1,4\}$ | $S_{14}$ | 3 |

Table 2: Examples of using the Oracle

**$(n_1, n_2)$-efficiency Oracle.** Consider any $\mathcal{B}$ denoting the set of base arms that the algorithm plans to learn the order (called *ordered base arms*). The Oracle can figure out the least set of super arms that are needed to learn the order (called *explored super arms*), and also the number of super arms that can be directly eliminated according to the learned order of $\mathcal{B}$ (called *eliminated super arms*). We call that $\mathcal{B}$ is $(n_1, n_2)$-efficiency when the number of "explored super arms" is $n_1$ while the number of "eliminated super arms" is $n_2$.

Intuitively, $n_1$ and $n_2$ measures the learning efficiency if the algorithm decides to learn the order of the base arms in $B$. The less the $n_1$, and the larger the $n_2$, the more quickly the algorithm can eliminate super arms.

Consider the SD-CMAB instance given by Table 1, Table 2 gives several examples on how our Oracle works. Here we explain the first example in Table 2, and other examples are just similar. According to the ordered base arms $\mathcal{B} = \{1,2,3\}$, the Oracle finds super arm $S_5$ covers all these base arms, and recognizes it as the explored super arm. As for the eliminated super arms, assuming we have already learned the order among base arm $1, 2$ and $3$. Then for super arms $S_1, S_2$ and $S_3$, since any two of them differs no more than 1 base arm, the order among these 3 base arms can definitely help identify two suboptimal super arms. This situation is the same for super arms $S_8$ and $S_{10}$. Therefore, the total number of eliminated super arms is 3.

We propose our algorithm, SortUCB-SD in Algorithm 3. Our algorithm performs in round basis. In round $h$, Oracle decides the set of "ordered base arms" denoted by $\mathcal{B}_h$, as well as the set of "explored super arms" denoted by $\mathcal{R}_h = \{R_{1,h}, R_{2,h,\ldots}\}$ according to the given input $\alpha_h$ and $\beta_h$. By uniformly pulling the "explored super arms", the algorithm learns the order of each base arm in $\mathcal{B}_h$, and the round ends whenever the orders of all base arms in $\mathcal{B}_h$ have been learnt.

---

**Algorithm 3** Sorting Upper Confidence Bound - Set Dependent

1: **Initialization:** $h \leftarrow 1, \mathcal{G} \leftarrow \mathcal{S}$
2: \\Sorting Phase        ▷ Find base arm sets $\mathcal{B}_h$ and super arm sets $R_{i,h}$
3: **Input:** $\alpha_1 > 0, \beta_1 > 0$.
4: **while** Oracle finds an $(n_1, n_2)$-efficiency ordered base arm set $\mathcal{B}_h$ with $n_1 \leq \alpha_h$, $n_2 \geq \beta_h$ **do**
5:    Supposing $R_{1,h}, R_{2,h}, \ldots$ denote the explored super arms
6:    **while** $\mathcal{B}_h$ does not satisfy (28) **do**
7:     Uniformly pulling super arm $R_{i,h}$.
8:     Update $N_{i(k),t}, \hat{\mu}_{i(k),t}$ and $t$
9:    **end while**
10:    Delete super arm $S \in \mathcal{G}$ according to Elimination Law (29)
11:    $h \leftarrow h + 1$, input new $\alpha_h$ and $\beta_h$.
12: **end while**
13: $\hat{\mu}_{i(k),N_{i(k),t}} \leftarrow 0$ for all $i \in [M]$ and $k \in [K]$, $N_{S,t} \leftarrow 0$ for all super arms $S \in \mathcal{G}$
14: \\UCB Phase        ▷ Using UCB to select the near-optimal super arm
15: **while** $t \leq T$ **do**
16:    Pull super arm with the highest (30) for the rest super arms $S \in \mathcal{G}$
17:    Update $N_{S,t}, \hat{\mu}_{i(k),t}$ and $t$
18: **end while**

---

Specifically, all the orders of the base arms in a set $\mathcal{B}_h$ have been learnt if for any $i, j \in \mathcal{B}_h$, it holds that

$$|\hat{\mu}_{i(k_{i,j}),N_{i(k_{i,j}),t}} - \hat{\mu}_{j(k_{i,j}),N_{j(k_{i,j}),t}}| \geq \sqrt{2 \log T}\left(\frac{1}{\sqrt{N_{i(k_{i,j}),t}}} + \frac{1}{\sqrt{N_{j(k_{i,j}),t}}}\right) \tag{28}$$

where $k_{i,j} \in [K]$ denotes some class. After the total $H$ rounds, the algorithm finishes the period of learning the order and remove a large amount of suboptimal super arms according to the "Elimination Law" which is introduced below.

**Elimination Law.** For some super arm $S$, if there exists $S' \in \mathcal{S}, S' \neq S, |S| \leq |S'|$, and there exist $i_1, i_2, \ldots, i_{|S|}$ and $j_1, j_2, \ldots, j_{|S|}, i_p \in [|S|], j_p \in [|S'|], i_p \neq i_{p'}$ when $p \neq p'$ s.t.

$$\hat{\mu}_{j_p(k_p), N_{j_p(k_p),t}} - \sqrt{\frac{2\log(T)}{N_{j_p(k_p),t}}} \geq \hat{\mu}_{i_p(k_p), N_{i_p(k_p),t}} + \sqrt{\frac{2\log(T)}{N_{i_p(k_p),t}}} \tag{29}$$

holds for all $p \in \{1, 2, \ldots, |S|\}$ and some $k_p \in [K]$ (which is associated with $p$), then eliminate super arm $S$.

For some problem instances, this procedure can be very effective since we can only use information of the reward of several base arms to eliminate a large number of super arms. The above manner can speed up the exploration process since we can avoid pulling many super arms which are apparently not the optimal super arm and thus lowers down regret and avoid exploring repeatedly.

Afterwards, we first reset all the estimation of base arms $\hat{\mu}_{i(k)}$ so that we can continue using an extension version of algorithm UCB (which consider each super arm as a single super arm and needs to record the sampled times for each super arm rather than each base arm) we consider each of the rest super arms as a single arm, and use an extension version of UCB algorithm to select the near-optimal super arm. That is, we just need to pull super arm $S$ with the highest

$$\sum_{i \in S} \hat{\mu}_{i(k_S), N_{i(k_S),t}} + \sqrt{\frac{2|S|\log T}{N_{S,t}}}. \tag{30}$$

The term $N_{S,t}$, similar to $N_{i(k),t}$, is used to denote the pulled times for super arm $S$ in the first $t$ time slots.

## G  PROOF OF THEOREM 3

*Proof of Theorem 3.* In algorithm 3 there exist two phases. We use $\text{Reg}_{1,T}(\pi, \nu)$ to denote regret in the Sorting Phase and $\text{Reg}_{2,T}(\pi, \nu)$ to denote regret in the UCB Phase. Thus

$$\text{Reg}_T(\pi, \nu) = \text{Reg}_{1,T}(\pi, \nu) + \text{Reg}_{2,T}(\pi, \nu)$$

As we totally use the Oracle for $H$ times in the first cycle, we have

$$\text{Reg}_{1,T}(\pi, \nu) = \sum_{h=1}^{H} \text{Reg}_{1(h),T}(\pi, \nu)$$

where $\text{Reg}_{1(h),T}(\pi, \nu)$ indicates the sum regret produced with the $h$-th use of Oracle. We use $T_h$ to indicate the time slots until the order of the 'explored super arms' returned by the $h$-th use of Oracle has been learned thoroughly, where $h \in [H]$.

W.L.O.G., we consider the $h$-th use of the Oracle. We define $G_{i(k),t}$ as follows:

$$G_{i(k),t} = \left\{ \hat{\mu}_{i(k), N_{i(k),t}} \in \left[ \mu_{i(k)} - \sqrt{\frac{2\log T}{N_{i(k),t}}}, \mu_{i(k)} + \sqrt{\frac{2\log T}{N_{i(k),t}}} \right] \right\}, \tag{31}$$

which shows the estimate of reward for arm $i$ with expectation $k$ in the $t$-th time slot is bounded in a range. The term $G_{i(k),t}^c$ indicates the complement event. We use $\mathbb{P}\left(G_{i(k),t}\right)$ to show the probability that event $G_{i(k),t}$ happens. Thus, $\mathbb{P}\left(G_{i(k),t}\right) + \mathbb{P}\left(G_{i(k),t}^c\right) = 1$.

Similar to $N_{i(k),t}$, we use $N_{S,t}$ to denote the times that super arm $S$ is pulled in the first $t$ time slots. We use $\tilde{N}_{S,T_h} = N_{S,T_h} - N_{S,T_{h-1}}$ to indicate the number of pulls for super arm $S$ between the $h$-th and $(h+1)$-th use of Oracle. Term $\tilde{N}_{S,t}^G$ denotes the pulling times for super arm $S$ in the first

$t$ slots with event $G = \bigcap_{i \in [M]} \bigcap_{K \in [K]} \bigcap_{t \in [T]} G_{i(\ell),t}$ happening, while $\tilde{N}_{S,t}^{G^c}$ denotes the opposite. According to 1,

$$\text{Reg}_{1(h),T}(\pi, \nu) = \sum_{S \in \mathcal{R}_h} \mathbb{E}\left[\tilde{N}_{S,T_h}\right] \Delta_S. \tag{32}$$

Our goal is to calculate $\mathbb{E}[\tilde{N}_{S,T_h}]$ for all super arms $S \in \mathcal{R}_h$. Obviously,

$$\mathbb{E}\left[\tilde{N}_{S,T_h}\right] = \mathbb{E}\left[\mathbb{I}\{G\}\,\tilde{N}_{S,T_h}^G\right] + \mathbb{E}\left[\mathbb{I}\{G^c\}\,\tilde{N}_{S,T_h}^{G^c}\right]. \tag{33}$$

As $\mathbb{I}\{G\} \le 1$ and $\tilde{N}_{S,T_h}^{G^c} \le T_h \le T$, combining (33), the upper bound for $\mathbb{E}\left[\tilde{N}_{S,T_h}\right]$ can be shown as follows

$$\mathbb{E}\left[\tilde{N}_{S,T_h}\right] \le \mathbb{E}\left[\tilde{N}_{S,T_h}^G\right] + T \cdot \mathbb{E}\left[\mathbb{I}\{G^c\}\right] = \mathbb{E}\left[\tilde{N}_{S,T_h}^G\right] + T \cdot \mathbb{P}\left(G^c\right).$$

Using Lemma 8, we obtain

$$\mathbb{P}\left(G^c\right) \le \frac{2MK}{T^3}, \ \mathbb{E}\left[\tilde{N}_{S,T_h}\right] \le \mathbb{E}\left[\tilde{N}_{S,T_h}^G\right] + \frac{2MK}{T^2}.$$

Below we consider the bound for $\mathbb{E}\left[\tilde{N}_{S,T_h}^G\right]$. In this cycle, our aim is to learn the order of base arms in base arm set $\mathcal{B}_h$. Assume the estimations of base arms in $\mathcal{B}_h$ have not reached the elimination condition (28). That is, there exists at least two base arms $i, j \in \mathcal{B}_h$, and for all $k \in [K]$,

$$\left|\hat{\mu}_{i(k),N_{i(k),t}^G} - \hat{\mu}_{j(k),N_{j(k),t}^G}\right| < \sqrt{2\log T}\left(\frac{1}{\sqrt{N_{i(k),t}^G}} + \frac{1}{\sqrt{N_{j(k),t}^G}}\right).$$

W.L.O.G, we assume $\hat{\mu}_{i(k),N_{i(k),t}^G} \ge \hat{\mu}_{j(k),N_{j(k),t}^G}$. Thus, in the time slot $t$, we have:

$$\hat{\mu}_{i(k),N_{i(k),t}^G} - \hat{\mu}_{j(k),N_{j(k),t}^G} < \sqrt{2\log T}\left(\frac{1}{\sqrt{N_{i(k),t}^G}} + \frac{1}{\sqrt{N_{j(k),t}^G}}\right)$$

for some $i, j \in [M]$ and all $k \in [K]$. With the definition of (33), we can derive that

$$\mu_{i(k)} - \mu_{j(k)} < 2\sqrt{2\log T}\left(\frac{1}{\sqrt{N_{i(k),t}^G}} + \frac{1}{\sqrt{N_{j(k),t}^G}}\right).$$

According to the definition of Oracle, super arm set $\mathcal{R}_h$ must cover any base arm $i \in \mathcal{B}_h$ (otherwise it is impossible to learn the order for all base arms in $\mathcal{B}_h$ with super arms in $\mathcal{R}_h$). As we uniformly pull super arms in $\mathcal{R}_h$, after every $|\mathcal{R}_h|$ time slots, each base arm $i \in \mathcal{B}_h$ can be achieved at least one time. That means $N_{i(k),t}^G - \tilde{N}_{S,t}^G \ge -1$ for any $S \in \mathcal{R}_h$. Thus, we can bound $\tilde{N}_{S,t}^G$ and as follows:

$$\tilde{N}_{S,t}^G < \frac{32\log T}{\Delta_{i(k),j(k)}^2} + 1$$

for all $t \in [T_{h-1} + 1, T_h]$, meaning $\tilde{N}_{S,T_h}^G < \frac{32\log T}{\Delta_{i(k),j(k)}^2} + 1$ holds. This means for any two base arms $i, j \in \mathcal{B}_h$, if there exists $S_1, S_2 \in \mathcal{R}_h, i \in S_1, j \in S_2, k_{S_1} = k_{S_2} = k$ satisfying the condition in the 14-th row in algorithm 3, we can eliminate set $B$. Thus, for the two base arms $i, j$, we should pull all the super arms in $\mathcal{R}_h$ for at least $\frac{32\log T}{\max_{\substack{S_1, S_2 \in \mathcal{R}_h \\ i \in S_1, j \in S_2 \\ k_{S_1} = k_{S_2} = k}} \Delta_{i(k),j(k)}^2} + 1$ time slots to ensure learning their order.

As we need to eliminate all the base arm sets in $\mathcal{B}$, the maximum pulling times for all super arms in $\mathcal{R}_h$ are

$$\mathbb{E}\left[\tilde{N}^G_{S,T_h}\right] \leq \frac{32\log T}{\min_{i,j\in\mathcal{B}_h}\max_{\substack{S_1,S_2\in\mathcal{R}_h\\i\in S_1,j\in S_2\\k_{S_1}=k_{S_2}=k}}\Delta^2_{i(k),j(k)}} + 1. \tag{34}$$

With this upper bound and proof above, we get $\mathbb{E}\left[[\tilde{N}_{S,T_h}\right] \leq \frac{32\log T}{\min_{i,j\in\mathcal{B}_h}\max_{\substack{S_1,S_2\in\mathcal{R}_h\\i\in S_1,j\in S_2\\k_{S_1}=k_{S_2}=k}}\Delta^2_{i(k),j(k)}} +$

$\frac{2MK}{T^2} + 1$. Thus, according to (32) and (34),

$$\text{Reg}_{1(h),T}(\pi,\nu) = \sum_{S\in\mathcal{R}_h}\mathbb{E}\left[\tilde{N}_{S,T_h}\right]\Delta_S \leq \left(\frac{32\log T}{\min_{i,j\in B}\max_{\substack{S_1,S_2\in\mathcal{R}_h\\i\in S_1,j\in S_2\\k_{S_1}=k_{S_2}=k}}\Delta^2_{i(k),j(k)}} + \frac{2MK}{T^2} + 1\right)\sum_{S\in\mathcal{R}_h}\Delta_S$$

$$= \left(\frac{32\log T}{\Delta^2_{\mathcal{B}_h,\min}} + \frac{2MK}{T^2} + 1\right)\sum_{S\in\mathcal{R}_h}\Delta_S.$$

Summing up for all $h\in[H]$, we have

$$\text{Reg}_{1,T}(\pi,\nu) \leq \sum_{h=1}^H\left(\frac{32\log T}{\Delta^2_{\mathcal{B}_h,\min}} + \frac{2MK}{T^2} + 1\right)\sum_{S\in\mathcal{R}_h}\Delta_S.$$

Here we end the proof of the bound for $\text{Reg}_{1,T}(\pi,\nu)$. Below we prove the bound for $\text{Reg}_{2,T}(\pi,\nu)$.

In this phase, our intuition is seeing each super arm as a single arm. We use $\mu_S = \sum_{i\in S}\mu_{i(k_S)}$ to denote the reward expectation for super arm $S$ and $\hat{\mu}_{S,N_{S,t}}$ to denote the unbiased estimate for super arm $S$ in the first $t$ time slots, while $N_{S,t}$ indicates the chosen times for super arm $S$ as a whole since the start of the UCB Phase, which is initialized to zero. We have

$$\text{Reg}_{2,T}(\pi,\nu) = \sum_{S\in\mathcal{G}}\mathbb{E}\left[N_{S,T}\right]\Delta_S. \tag{35}$$

First we give a lemma that ensures the optimal super arm can be in $\mathcal{G}$ with high probability.

**Lemma 17.** *If event $G$ happens, the optimal super arm (denoted by $S^*$) must be concluded in $\mathcal{G}$.*

*Proof of lemma 17.* As we defined, event $G$ means $\mu_{i(k)} - \sqrt{\frac{2\log T}{N_{i(k),t}}} \leq \hat{\mu}_{i(k),N_{i(k),t}} \leq \mu_{i(k)} + \sqrt{\frac{2\log T}{N_{i(k),t}}}$ is true for each $i\in[M], k\in[K], t\in[T]$. We just need to prove what we eliminate in line 8 in Algorithm 3 are suboptimal super arms.

According to the elimination condition (29), when there exists $i_1, i_2, \ldots, i_{|S|}$ and $j_1, j_2, \ldots, j_{|S|}$ s.t.

$$\hat{\mu}_{j_p(k_p),N_{j_p(k_p),t}} - \sqrt{\frac{2\log T}{N_{j_p(k_p),t}}} \geq \hat{\mu}_{i_p(k_p),N_{i_p(k_p),t}} + \sqrt{\frac{2\log T}{N_{i_p(k_p),t}}}, \tag{36}$$

we can eliminate super arm $S$. Combining the definition of (31), inequalities $\hat{\mu}_{i_p(k_p),N_{i_p(k_p),t}} + \sqrt{\frac{2\log T}{N_{i_p(k_p),t}}} \geq \mu_{i_p(k_p)}$ and $\hat{\mu}_{j_p(k_p),N_{j_p(k_p),t}} - \sqrt{\frac{2\log T}{N_{j_p(k_p),t}}} \leq \mu_{j_p(k_p)}$ hold. Take (36) into consideration, that is $\mu_{j_p(k_p)} \geq \mu_{i_p(k_p)}$, meaning base arm $j_p$ is better than $i_p$ under $G_{i(k),t}$.

As this quality holds for all base arms in $S$, this means for any base arm $i$ in $S$, there exists a different base arm in $S'$ which is better than $i$. Thus,

$$\mu_S = \sum_{i\in S}\mu_{i(k_S)} \leq \sum_{i\in S'}\mu_{i(k_{S'})} = \mu_{S'} \tag{37}$$

shows super arm $S$ is a suboptimal arm. Therefore, any eliminated arm in this phase must be a suboptimal arm when event $G$ happens. Here we end the proof of lemma 17. $\square$

Now we continue our proof of $\text{Reg}_{2,T}(\pi,\nu)$. As we have proved in previous,

$$\text{Reg}_{2,T}(\pi,\nu) = \sum_{S\in\mathcal{G}} \mathbb{E}\left[N_{S,T}\right]\Delta_S \leq \sum_{S\in\mathcal{G}} \mathbb{E}\left[N_{S,T}^G\right]\Delta_S + \frac{2MK}{T^2}\sum_{S\in\mathcal{G}}\Delta_S$$

$$= \sum_{\substack{S\in\mathcal{G}\\S\neq S^*}} \mathbb{E}\left[N_{S,T}^G\right]\Delta_S + \frac{2MK}{T^2}\sum_{S\in\mathcal{G}}\Delta_S. \tag{38}$$

The last equation is because of $\Delta_{S^*} = 0$. Thus, we only need to prove the upper bound for $\mathbb{E}[N_{S,T}^G]$. We first define another event $\tilde{G}_S$ which we need in our proof:

$$\tilde{G}_S = \left\{\mu_{S^*} < \min_{t\in[T]}\hat{\mu}_{S^*,N_{S^*,t}} + \sqrt{\frac{2|S^*|\log T}{N_{S^*,t}}}\right\} \cap \left\{\hat{\mu}_{S,u_S} + \sqrt{\frac{2|S|\log T}{u_S}} < \mu_{S^*}\right\}, \tag{39}$$

where $u_S \in [T]$ is a constant to be chosen later.

Below we give two lemmas.

**Lemma 18.** *If $\tilde{G}_S$ occurs, then $N_{S,T} \leq u_S$.*

**Lemma 19.** *$\tilde{G}_S^c$, meaning the complement part of $\tilde{G}_S$, happens with low probability.*

As $N_{S,T}^G \leq T$, we use $N_{S,t}^{G\cap\tilde{G}_S}$ to denote the pulling times for super arm $S$ in the first $t$ time slots in the second cycle with event $G$ and $\tilde{G}_S$ both happening, while $N_{S,t}^{G\cap\tilde{G}_S^c}$ means that only event $G$ happens while event $\tilde{G}_S$ does not happen. Then,

$$\mathbb{E}\left[N_{S,T}^G\right] = \mathbb{E}\left[\mathbb{I}(\tilde{G}_S)N_{S,T}^{G\cap\tilde{G}_S}\right] + \mathbb{E}\left[\mathbb{I}(\tilde{G}_S^c)N_{S,T}^{G\cap\tilde{G}_S^c}\right] \leq \mathbb{E}\left[N_{S,T}^{G\cap\tilde{G}_S}\right] + T\cdot\mathbb{P}\left(\tilde{G}_S^c\right). \tag{40}$$

*Proof of lemma 18.* Assuming that $\tilde{G}_S$ occurs with $N_{S,T} \geq u_S$. That means there exists $t \in T$ s.t. $N_{S,t-1} = u_S$ while $A_t = S$, where $A_t$ means the chosen super arm at time slot $t$. According to (39), we have:

$$\hat{\mu}_{S,N_{S,t-1}} + \sqrt{\frac{2|S|\log T}{N_{S,t-1}}} \leq \mu_{S^*} \leq \hat{\mu}_{S^*,N_{S,t-1}} + \sqrt{\frac{2|S^*|\log T}{N_{S,t-1}}}. \tag{41}$$

That means in time slot $t$ we should choose super arm $S^*$ rather than arm $S$, which is a contradiction. $\square$

*Proof of lemma 19.* The complement part of $\tilde{G}_S$ is

$$\tilde{G}_S^c = \left\{\mu_{S^*} \geq \min_{t\in[T]}\left(\hat{\mu}_{S^*,N_{S^*,t}} + \sqrt{\frac{2|S^*|\log T}{N_{S^*,t}}}\right)\right\} \cup \left\{\hat{\mu}_{S,u_S} + \sqrt{\frac{2|S|\log T}{u_S}} \geq \mu_{S^*}\right\}. \tag{42}$$

We first prove the first part of $\tilde{G}_S^c$. As

$$\left\{\mu_{S^*} \geq \min_{t\in[T]}\left(\hat{\mu}_{S^*,t} + \sqrt{\frac{2|S^*|\log T}{N_{S^*,t}}}\right)\right\} \subset \left\{\mu_{S^*} \geq \min_{t\in[T]}\left(\hat{\mu}_{S^*,t} + \sqrt{\frac{2|S^*|\log T}{t}}\right)\right\}$$

$$= \bigcup_{t\in[T]}\left\{\mu_{S^*} \geq \hat{\mu}_{S^*,t} + \sqrt{\frac{2|S^*|\log T}{t}}\right\}.$$

Thus, using Lemma 5, we have:

$$\mathbb{P}\left(\mu_{S^*} \geq \min_{t \in [T]}\left(\hat{\mu}_{S^*,t} + \sqrt{\frac{2|S^*|\log T}{N_{S^*,t}}}\right)\right) \leq \mathbb{P}\left(\bigcup_{t \in [T]}\left\{\mu_{S^*} \geq \hat{\mu}_{S^*,t} + \sqrt{\frac{2|S^*|\log T}{t}}\right\}\right)$$

$$\leq \sum_{t=1}^{T} \mathbb{P}\left(\mu_{S^*} \geq \hat{\mu}_{S^*,t} + \sqrt{\frac{2|S^*|\log T}{t}}\right) \leq \frac{1}{T^3}, \tag{43}$$

which is a low probability if $T$ is chosen large enough. For the last inequality in (43), we use lemma 4 with $t|S^*|$ independent samples.

Next we bound the second part of $\tilde{G}_S^c$. As $u_S$ is a parameter undetermined, we assume it is large enough that $\Delta_S - \sqrt{\frac{2|S|\log T}{u_S}} \geq c\Delta_S$, where $c \in (0,1)$ will be chosen later. Thus,

$$\mathbb{P}\left(\hat{\mu}_{S,u_S} + \sqrt{\frac{2|S|\log T}{u_S}} \geq \mu_{S^*}\right) = \mathbb{P}\left(\hat{\mu}_{S,u_S} - \mu_S \geq \Delta_S - \sqrt{\frac{2|S|\log T}{u_S}}\right)$$

$$\leq \mathbb{P}\left(\hat{\mu}_{S,u_S} - \mu_S \geq c\Delta_S\right) \leq \exp\left(-2c^2\Delta_S^2 \frac{u_S}{|S|}\right). \tag{44}$$

Taking together (43) and (44),

$$\mathbb{P}\left(\tilde{G}_S^c\right) \leq \frac{1}{T^3} + \exp\left(-2c^2\Delta_S^2\frac{u_S}{|S|}\right).$$

Here we end the proof of lemma 19. $\qquad\square$

As we have proved in (40)

$$\mathbb{E}\left[N_{S,T}^G\right] \leq u_S + T\left(\frac{1}{T^3} + \exp\left(-2c^2\Delta_S^2\frac{u_S}{|S|}\right)\right) = u_S + T\exp\left(-2c^2\Delta_S^2\frac{u_S}{|S|}\right) + \frac{1}{T^2}.$$

Choosing $u_S = \lceil \frac{2|S|\log T}{(1-c)^2\Delta_S^2}\rceil$ and $c = 1/2$, then

$$\mathbb{E}[N_{S,T}] \leq 3 + \frac{8|S|\log T}{\Delta_S^2}.$$

Considering (38), we can give an upper bound for $\text{Reg}_{2,T}(\pi,\nu)$,

$$\text{Reg}_{2,T}\left(\pi,\nu\right) \leq \sum_{S \in \mathcal{G}, S \neq S^*}\left(3\Delta_S + \frac{8|S|\log T}{\Delta_S}\right) + \frac{MK}{T^2}\sum_{S \in \mathcal{G}}\Delta_S. \tag{45}$$

Then we take together (34) and (45) and get

$$\text{Reg}_T(\pi,\nu) \leq \sum_{h=1}^{H}\left(\frac{32\log T}{\Delta_{\mathcal{B}_h,\min}^2}\sum_{S \in \mathcal{R}_h}\Delta_S\right) + \sum_{S \in \mathcal{G}, S \neq S^*}\frac{8|S|\log T}{\Delta_S}$$

$$+ \left(\frac{2MK}{T^2} + 3\right)\left(\sum_{h=1}^{H}\sum_{S \in \mathcal{R}_h}\Delta_S + \sum_{S \in \mathcal{G}, S \neq S^*}\Delta_S\right). \tag{46}$$

Theorem 3 is the simplified version of (46). $\qquad\square$

