# OpenReview forum: "Set-Size Dependent Combinatorial Bandits"
_ICLR.cc/2025/Conference — Submitted to ICLR 2025_

### Official Review · Reviewer_C6dM · 2024-10-23

**Soundness:** 2
**Presentation:** 2
**Contribution:** 2
**Rating:** 3
**Confidence:** 4

**Summary:**

The authors introduce and study Set-Size Dependent Combinatorial Multi-Armed Bandits (SSD-CMAB) where each base arm is associated with a set of different reward distributions, and the reward distribution of each base arm depends on the set size. The authors propose the SortUCB algorithm, leveraging the order preservation property to reduce the exploration space and provide theoretical upper bound regret guarantees. Moreover, a lower bound is derived showing that SortUCB is relatively tight. Finally the authors conduct some numerical experiments, showing good performance of SortUCB.

**Strengths:**

The authors derive a lower bound for the studied problem.

The authors derive a regret upper bound for the proposed SortUCB algorithm.

The authors conduct some numerical analysis of the proposed approach.

**Weaknesses:**

**Missing discussion/contrast/comparison with closely related works:**

It is fine that a related work section is put in the appendix. However, the main paper should cite related works at least briefly and then refer for detailed discussion in the appendix. Hence, closely related topics should not be omitted in the main paper.

A very closely related research to this work are submodular bandits, which are a sort of CMAB, yet with submodular (more general) rewards. In such settings, the reward of arms indeed depends on the set (including its size and more, hence generalizing their setting). While some of these works were mentioned briefly in the Appendix A. We believe, due to its high relevance to their considered setting, it must be clearly discussed and highlighted in the paper with key differences and motivations.

Several works have been published tackling the CMAB with general rewards including semi-bandit (like this work) [1, 2] and even for (more general) bandit feedback [3]. These have to be discussed and the work should contrast their results to these works (theoretically and/or empirically).

The authors should compare their work (at least empirically) to recent works on CMAB where arms depend on the set, such as submodular bandits. For example, given the consideration of a fixed cardinality constraint, recent algorithms, like [3] can be considered for comparison.

[1] Chen, L., Harshaw, C., Hassani, H., & Karbasi, A.. (2018). Projection-Free Online Optimization with Stochastic Gradient: From Convexity to Submodularity. In Proceedings of the ICML.

[2] Takemori, S., Sato, M., Sonoda, T., Singh, J., & Ohkuma, T. (2020). Submodular bandit problem under multiple constraints. In Proceedings of the UAI.

[3] Fourati, F., Quinn, C. J., Alouini, M. S., & Aggarwal, V. (2024). Combinatorial stochastic-greedy bandit. In Proceedings of the AAAI.

**Motivation:**

The proposed problem can be solved using the same CMAB frameworks which estimates the super arm reward directly but considering LM arms instead of M arms. While the problem shows a linear increase with increased constraint (L). The increase remains linear and in general L is not large. In recommender systems such a constraint is usually much smaller than M (L<<M). Furthermore, submodular bandits can be used to tackled these variable distributions base on the set. Hence, the motivation of the work remains unclear.

**Approach limitations:**

Assume a linear reward function and semi-bandit feedback, unlike other works which tackles non-linear and/or bandit feedback, which limits the applicability of these approaches. (Minor: the authors should be clear about this earlier in the paper, possibly from the introduction).

**Complexity Analysis Missing:**

The authors do not provide a time and space complexity analysis (even though the algorithm has three different phases, each requiring different complexities, and in some of these requires several comparisons with other arms and super-arms).

**Numerical Analysis Limitations:**

Compare only with two benchmarks. CMAB (2015) and MPMAB (1985). Several combinatorial bandit algorithms have been suggested which includes their proposed setting as a special case, such as submodular bandits (mentioned in the above discussion).

The number of arms remains limited. L=8 is fine, but larger M should be considered for more realistic comparison.

Diverse synthetic and real-world datasets should be considered. With limited datasets/settings and a few considered methods it is very hard to justify the practicality and good empirical performance of the method.


**Minors:**

There is an error in the abstract for the regret upper bound. Should be delta_L^2 based on their proposed theoretical results.

Algorithm 1, line 11, mentions delete any base arm. However, it is unclear from where, R or B? I assume from R.

**Questions:**

How does the SortUCB compare, in practice, against submodular bandit works, given that they consider a variable arm distribution when selected within different sets?

The remainder of the algorithm (UCB Phase) focuses solely on exploitation within the set A of L main arms. It is not clear why the considered subsets of A are only L^2 and not 2^L? The subsets that can be composed with L arms are 2^L. If so, the regret should be exponential with L and not quadratic!

---

> ### Author Response · Authors · 2024-11-22
>
> Thank you for your valuable advices!
>
> ### 1. For the concern about related works:
>
> #### (i). For Your Concern About the Suitable Place of Related Work:
>
> We have revised the Introduction section to include discussions of many of the most related works, particularly in paragraph 3.
> Additionally, we have enriched the Related Work section in Appendix A to provide a more comprehensive overview.
>
>
> #### (ii). For Your Concern About the Relationship Between Our Setting and CMAB with Submodular Functions:
>
> We have emphasized this relationship in both the 3rd paragraph of the Introduction and the 2nd paragraph of the Related Works in Appendix A.
> In our rebuttal revision, we have included all the papers referred to in your review.
> Additionally, we have added discussions in Appendix A to highlight the key differences and motivations for our setting.
>
>
> #### (iii). For Your Concern About Comparing Our Setting with "CMAB with Submodular Reward Functions" Theoretically and Empirically:
>
> We would like to clarify that, although there are some similarities between the two settings, our setting (SSD-CMAB) does not assume the most important properties of submodular functions, i.e., monotonicity and submodularity. Specifically, the reward function for a super arm in CMAB with submodular functions is required to satisfy:
> $$
> r(S \cup \{i\}) \geq r(S) \quad \text{and} \quad r(S \cup \{i\}) - r(S) \geq r(S' \cup \{i\}) - r(S'),
> $$
> for all base arms $i$ and super arms $S, S'$ such that $S \subseteq S'$.
>
> In contrast, the only assumption in our model is the order-preservation property for the reward means of base arms, which is significantly different from the submodular function. To illustrate this distinction, consider an SSD-CMAB instance with $M=L=4$. Let the reward mean for arm $i$ in a super arm with size $\ell$ be denoted by $\mu_{i(\ell)}$, where $i \in [M]$ and $\ell \in [L]$. The reward matrix for this instance is as follows:
>
> |          | $i=1$    |  $i=2$   |  $i=3$   |   $i=4$  |
> |----------|----------|----------|----------|----------|
> | $\ell=1$ | $\mu_{1(1)}=0.9$  | $\mu_{2(1)}=0.8$ | $\mu_{3(1)}=0.7$  | $\mu_{4(1)}=0.6$
> | $\ell=2$ | $\mu_{1(2)}=0.8$  | $\mu_{2(2)}=0.6$ | $\mu_{3(2)}=0.5$  | $\mu_{4(2)}=0.4$
> | $\ell=3$ | $\mu_{1(3)}=0.9$  | $\mu_{2(3)}=0.8$ | $\mu_{3(3)}=0.4$  | $\mu_{4(3)}=0.05$
> | $\ell=4$ | $\mu_{1(4)}=0.5$  | $\mu_{2(4)}=0.3$ | $\mu_{3(4)}=0.2$  | $\mu_{4(4)}=0.1$
>
> This instance satisfies the order-preservation property and is therefore valid under our setting. However, the reward expectation of each base arm may increase or decrease with the set size. For example, the reward mean of base arm 1 decreases from 0.9 to 0.8 when the size grows from 1 to 2, but increases again to 0.9 when the size grows to 3. Using the terminology of submodular functions, we find that:
> $$
> r(\{1,2,3\} \cup \{4\}) < r(\{1,2,3\}) \quad \text{and} \quad 0.5 = r(\{1\} \cup \{2\}) - r(\{1\}) < r(\{1,3\} \cup \{2\}) - r(\{1,3\}) = 0.8.
> $$
> In this case, the reward function does not satisfy monotonicity or submodularity. Thus, while we can highlight the differences between the two settings, a direct theoretical or empirical comparison is not feasible due to the differing properties and assumptions.

---

> > ### Author Response · Authors · 2024-11-22
> >
> > ### 2. For the Concern About Motivation:
> >
> > It is true that the SSD-CMAB problem can be addressed using CMAB frameworks by considering $LM$ arms instead of $M$ arms. However, we propose this as a new model because it corresponds to many real-life scenarios. We provide several motivations for this problem, including applications in advertising and utility maximization.
> >
> > In certain scenarios, $L$ can approximate $M$—for example, in advertising, where a single page can often accommodate many advertisements. Similarly, in the utility maximization problem introduced in the Introduction, the number of users sharing signals (denoted by $L$) can also be large.
> >
> > As previously mentioned, submodular bandits cannot capture many scenarios that our model can address. For instance, even in cases where the reward mean of each base arm decreases as the set size grows, our model does not satisfy the monotonicity property of submodular functions. Consider the following example:
> >
> > |          | $i=1$    |  $i=2$   |  $i=3$   |   $i=4$  |
> > |----------|----------|----------|----------|----------|
> > | $\ell=1$ | $\mu_{1(1)}=0.9$  | $\mu_{2(1)}=0.8$ | $\mu_{3(1)}=0.7$  | $\mu_{4(1)}=0.6$
> > | $\ell=2$ | $\mu_{1(2)}=0.8$  | $\mu_{2(2)}=0.6$ | $\mu_{3(2)}=0.5$  | $\mu_{4(2)}=0.4$
> > | $\ell=3$ | $\mu_{1(3)}=0.7$  | $\mu_{2(3)}=0.5$ | $\mu_{3(3)}=0.4$  | $\mu_{4(3)}=0.3$
> > | $\ell=4$ | $\mu_{1(4)}=0.4$  | $\mu_{2(4)}=0.2$ | $\mu_{3(4)}=0.1$  | $\mu_{4(4)}=0.05$
> >
> > Here, the reward of each base arm decreases with the set size. However, for the super arm $S=\{1,2,3\}$ and base arm $a$, we have
> > $$
> > 0.75 = r(S \cup \{a\}) < r(S) = 1.6,
> > $$
> > which contradicts the monotonicity property of submodular functions.
> >
> > While modeling motivations such as advertising and utility maximization as SSD-CMAB is natural, submodular bandits can only capture part of these motivations, not all of them. Thus, we believe that our model is well-suited and reasonable for the motivations we have discussed.
> >
> >
> > ### 3. For the Concern About Approach Limitations:
> >
> > Thank you for your suggestions. We have revised paragraph 3 in the Introduction to include a discussion of the linear reward function and semi-bandit feedback in our model. Regarding your concerns about general reward functions and bandit feedback, these are indeed valuable directions for future research but are not the focus of this paper.
> >
> > As far as we know, many existing studies also focus on linear reward functions and semi-bandit feedback. For example:  "Kveton, B., Wen, Z., Ashkan, A., & Szepesvari, C. (2015, February). Tight regret bounds for stochastic combinatorial semi-bandits."  Moreover, since this is the first paper introducing this model, we believe that it is reasonable to consider these foundational cases initially.
> >
> >
> > ### 4. For the concern about Complexity Analysis Missing:
> >
> > Thank you for your vital advices. We have added the complexity analysis in the end of Section 3 and describe which in detail in Appendix C. Please let us know whether you have other problems about the implementation.
> >
> > ### 5. For the Concern About Numerical Analysis Limitations:
> >
> > As we have clarified before, submodular bandits differ significantly from our model and cannot capture some of the scenarios that our model addresses. The reason we only consider benchmarks like CombUCB1 and MPMAB is that these are the only two algorithms we have identified that can be reduced to our model. Recent works in CMAB, such as submodular bandits, operate under different assumptions and focus on models that differ greatly from ours. Thus, we could not identify more reasonable algorithms for comparison.
> >
> > Regarding your suggestion to include additional events, we appreciate the constructive feedback and will carefully evaluate the inclusion of additional events to enhance the robustness and comprehensiveness of our numerical analysis. If deemed necessary, we plan to incorporate them in future revisions to further strengthen our study.

---

> ### Comment · Reviewer_C6dM · 2024-11-23
> **Submodular functions are not necessarily monotone, and empirical comparisons with submodular bandits are indeed feasible!**
>
> Thank you for your reply.
>
> The second key concern, apart from the position of the related works, is the lack of closely related works. The references mentioned in the rebuttal represent only a small subset of the extensive literature on submodular maximization. We appreciate adding those references. However, the authors are strongly encouraged to explore more.
>
> **The paper contains some incorrect statements:** (Line 674: However, submodular functions require two key properties, i.e., monotonicity and submodularity. … Hence, our model is able to capture scenarios that submodular bandits cannot capture, especially when the reward for some base arms does not change monotonically with a growing set size.) And even in the rebuttal, the authors mentions ((iii) ...our setting (SSD-CMAB) does not assume the most important properties of submodular functions, i.e., monotonicity and submodularity.)
>
> **Submodular functions do not require monotonicity** (though some algorithms' gurantees do, but not all). There is a significant body of research on non-monotonic submodular functions [4, 5, 6]. While the guarantees differ for non-monotonic submodular functions, it is misleading to claim that monotonicity is a necessary property of submodular functions. The authors must revise such statements and are encouraged to consult the references below and explore other recent closely related works:
>
> [4] Feige, U., Mirrokni, V. S., & Vondrák, J. (2011). Maximizing non-monotone submodular functions. SIAM Journal on Computing.
>
> [5] Niazadeh, R., Golrezaei, N., Wang, J. R., Susan, F., & Badanidiyuru, A. (2021, July). Online learning via offline greedy algorithms: Applications in market design and optimization. Proceedings of the 22nd ACM Conference on Economics and Computation.
>
> [6] Fourati, F., Aggarwal, V., Quinn, C., & Alouini, M. S. (2023, April). Randomized greedy learning for non-monotone stochastic submodular maximization under full-bandit feedback. International Conference on Artificial Intelligence and Statistics.
>
> **A second incorrect claim is that an empirical comparison with submodular bandits is not feasible.** While submodular bandit algorithms use the assumption of submodularity to derive theoretical guarantees, they do not require a submodular function to execute. In other words, submodularity is not a prerequisite for running the algorithms but rather for ensuring the guarantees. Therefore, an empirical comparison is certainly feasible and can provide valuable insights from an algorithmic perspective. The authors are strongly encouraged to consider some of the most recent submodular bandit algorithms for an empirical comparison, which could further strengthen the motivation for their studied settings.
>
> The submodular bandit literature (monotone and non-monotone) addresses most of the proposed motivations and represent a strong and highly relevant benchmark for this work, yet they are overlooked in both the empirical study and theoretical analysis.

---

> > ### Author Response · Authors · 2024-11-24
> >
> > Thank you for your prompt response and valuable advices.
> >
> > ### 1. For the concern about lacking of related literatures:
> > We have included more literatures in the Related Work part currently, including those you have
> > referred to in the rebuttal. We will keep researching more associated literatures in the future
> > and a version with more detailed related works is expected to be submitted before
> > the deadline.
> >
> > ### 2. For the concern about incorrect statements with the submodular functions:
> > Thank you for your correction and we apologize for including the misleading points. We have fixed the related points you referred to in the rebuttal revision.
> >
> > ### 3. For the concern about the relationship between submodular bandits and our setting:
> > We hope to clarify that our setting, SSD-CMAB with linear reward, is not a special case of submodular bandits
> > even for submodular bandits without the monotonicity property, since many cases in SSD-CMAB does not
> > assume and satisfy the submodular peoperty of reward functions. We give two cases below,
> > each of which represents one different scenario. We still suppose $M=L=4$ and use $\mu_{i(\ell)}$ to denote the mean reward for arm $i$ in super arms with size $\ell$:
> >
> > #### Case 1.
> > |          | $i=1$    |  $i=2$   |  $i=3$   |   $i=4$  |
> > |----------|----------|----------|----------|----------|
> > | $\ell=1$ | $\mu_{1(1)}=1$  | $\mu_{2(1)}=0.8$ | $\mu_{3(1)}=0.7$  | $\mu_{4(1)}=0.6$
> > | $\ell=2$ | $\mu_{1(2)}=0.75$  | $\mu_{2(2)}=0.65$ | $\mu_{3(2)}=0.55$  | $\mu_{4(2)}=0.5$
> > | $\ell=3$ | $\mu_{1(3)}=0.7$  | $\mu_{2(3)}=0.6$ | $\mu_{3(3)}=0.5$  | $\mu_{4(3)}=0.4$
> > | $\ell=4$ | $\mu_{1(4)}=0.6$  | $\mu_{2(4)}=0.5$ | $\mu_{3(4)}=0.4$  | $\mu_{4(4)}=0.3$
> >
> > This is the case that the reward mean of each base arm decreases with the set size, corresponding to some of our motivations mentioned in the Introduction part. However, recall the definition of submodular:
> > $$r(S\cup a)-r(S)\geq r(S'\cup a)-r(S')$$
> > for super arm $S\subseteq S'$. Let $S=\{1\}$, $a=\{3\}$, $S'=\{1,2\}$, then we have
> > $$0.3=0.75+0.55-1=r(\{1,3\})-r(\{1\})$$
> > $$< r(\{1,2,3\})-r(\{1,2\})=0.7+0.6+0.5-(0.75+0.65)=0.4$$
> > which is a contradictory to the submodular property.
> >
> > #### Case 2.
> > |          | $i=1$    |  $i=2$   |  $i=3$   |   $i=4$  |
> > |----------|----------|----------|----------|----------|
> > | $\ell=1$ | $\mu_{1(1)}=0.9$  | $\mu_{2(1)}=0.8$ | $\mu_{3(1)}=0.7$  | $\mu_{4(1)}=0.6$
> > | $\ell=2$ | $\mu_{1(2)}=0.8$  | $\mu_{2(2)}=0.6$ | $\mu_{3(2)}=0.5$  | $\mu_{4(2)}=0.4$
> > | $\ell=3$ | $\mu_{1(3)}=0.9$  | $\mu_{2(3)}=0.8$ | $\mu_{3(3)}=0.4$  | $\mu_{4(3)}=0.05$
> > | $\ell=4$ | $\mu_{1(4)}=0.5$  | $\mu_{2(4)}=0.3$ | $\mu_{3(4)}=0.2$  | $\mu_{4(4)}=0.1$
> >
> > This is the case that the reward mean of each base arm does not essentially decrease or increase with the set size, which can also be captured by our SSD-CMAB setting as it satisfies the order preservation property. It can be verified that this example contradicts the submodular property in submodular reward function (as the reward mean in this case can even sometimes increase with the set size).
> >
> > Generally, we believe that although submodular bandits share some similarities with our setting, they are indeed two different settings. While the core property in submodular bandits is submodular and that in our setting is order preservation, the submodular property cannot cover our order preservation property in many cases which correspond to our motivations. Therefore our setting is not just a special case of submodular bandits and can better explain the motivations in our paper.
> >
> > ### 4. For the concern about the empirical comparison with submodular bandits:
> > As we have claimed in the 3rd point, submodular bandits have different assumption with our setting, and cannot capture many scenarios that our setting can. We believe that using algorithms of submodular bandits in our setting may possibly lead to linear regret (or at least perform much worse than the other baselines such as CombUCB1). We will try to implement algorithms associated with submodular bandits and do experiments on our setting in the future to verify whether its performance is bad. And if we have some results before the deadline we would update which with you.

---

> ### Comment · Reviewer_C6dM · 2024-11-25
>
> Thank you for your reply. We appreciate your effort to improve the draft.
>
> **Related Works Problems:**
>
> While the authors removed the incorrect statement that submodularity requires monotonicity, we notice that some parts of the manuscript still implicitly use this misconception to argue the advantages of their work.
>
> line 731-734: "However, one common property that submodular function requires is the submodular property. In contrast, SSD-CMAB only requires order-preservation, **allowing base arm rewards to increase or decrease with super arm
> size**. Hence, our model is able to capture many scenarios that submodular bandits cannot capture,
> **especially when the reward for some base arms do not change monotonically with set size growing.**"
>
> The authors should emphasize their actual advantages over submodular bandits beyond addressing monotonicity, as submodular bandits also accommodate non-monotonicity.
>
> The authors in the main paper line 66-71: "Note that in the CMAB model the reward distribution of a base arm remains the same across all super arms. **In contrast,** SSD-CMAB models base arm rewards as dependent on the size of the super arm, with
> each base arm having L different reward distributions. **This is the main distinction with CMAB** where each base arm has only one fixed distribution (Gai et al., 2012; Combes et al., 2015b), **even in non-linear reward settings** (Chen et al., 2016b; 2021; Merlis & Mannor, 2019) **or works that focus on the arms depending on the set** (Chen et al., 2018; Takemori et al., 2020; Fourati et al., 2024).''
>
> This statement is somewhat misleading. While CMAB with submodular rewards does assume submodularity, it does not necessarily imply that the reward distribution of a base arm remains the same across all super arms, similarly to SSD-CMAB (no contrast here). The authors should clearly and explicitly articulate the differences between their work and prior studies, avoiding any ambiguous or misleading statements.
>
> **Empirical Comparison:**
>
> You may believe that using submodular bandit algorithms in your setting might lead to linear regret (or at least perform significantly worse than other baselines, such as CombUCB1). However, **an empirical comparison is essential to substantiate these beliefs.**
>
> We understand that your model has slightly different assumptions from those of submodular rewards, but this does not necessarily imply that submodular bandits will perform poorly under these slightly different assumptions (at least from an empirical perspective), especially that they have a lot in common (rewards being dependent on the set).
>
> Combinatorial bandits are simple and easy to implement and do not require any GPU or extensive memory to test. Given their high relevance to your setting (higher than CombUCB), there is no reason to delay such a comparison to future works.

---

> > ### Author Response · Authors · 2024-11-27
> >
> > Thank you for your suggestions. We would fix the problems you referred to in the related works, and a version with more explicit statements is expected to be submitted before the deadline.
> >
> > As for your concern about the comparison with submodular bandits, we will try to add experiments related to this setting. As we have not found any implemented code online on submodular bandits with semi-bandit feedback currently, we will try to implement one. If we can get enough results we will add these experiments in the future versions.

---

### Official Review · Reviewer_8cZM · 2024-10-24

**Soundness:** 2
**Presentation:** 2
**Contribution:** 2
**Rating:** 5
**Confidence:** 3

**Summary:**

This paper proposes a new framework of combinatorial multi-armed bandits named Set-Size Dependent combinatorial multi-armed bandit (SSD-CMAB). In SSD-CMAB, the reward distribution of each base arm depends on the size of the chosen super arm. As a key assumption, there is an order preservation, which means that the order of the reward means of base arms is independent of the set size. This work shows upper and lower bounds of regret and shows that their proposed algorithm, SortUCB, is partially tight. Finally, it numerically evaluates SortUCB.

**Strengths:**

The problem setting seems novel since the ordinary CMAB assumes that the rewards of each base arm do not depend on the super arm. The SSD-CMAB framework can model some online advertising where the click rate for each ad will decrease when a large number of ads are shown to a user at once.

**Weaknesses:**

- It seems that Algorithm 1 has some issues in computational complexity. $N_{S, t}$ maintains the number of times super arm $S$ has been selected. However, in general, the size of the set of super arms is exponentially large with respect to the number of base arms. Therefore, this algorithm potentially consumes an enormous amount of memory to maintain $\\{N_{S, t}\\}_{S = 1, \ldots, |\mathcal{A}|}$. In addition, finding the super arm with the highest value for (4) is a nonlinear optimization problem with combinatorial constraints, which is computationally heavy.
- In my understanding, combinatorial MAB means that there is a combinatorial structure in the set of super arms. However, SSD-CMAB defines a set of super arms as a subset of base arms whose cardinality is no more than a certain number. I am not sure if we can say this as a combinatorial structure.

**Questions:**

- Why is not the CTS algorithm from S. Wang et al. (ICML2018) compared with the SortUCB algorithm in the Experiments section? I believe this algorithm performs empirically well in semi-bandit problems.
- In Section 5 (about SD-CMAB), there is a notion of \emph{class} whose intuition is ambiguous. Taking the example of paths, what are the $K$ classes of paths?

---

> ### Author Response · Authors · 2024-11-22
>
> Thank you for your valuable advices. We address your concern of our paper as follows:
>
> ### 1. Concern about the computational complexity:
>
> Thank you for your vital advices. We have added the complexity analysis in the end of Section 3 and describe which in detail in Appendix C. Indeed the computational complexity per time slot is no more than $O\left(M\log(M)\right)$, which is a low computation cost. Please let us know whether you have other problems about the implementation.
>
> ### 2. Concern About the Combinatorial Structure:
>
> We introduce the term $L$ in our model because it is commonly used in prior works on CMAB, such as:
> - "Kveton, B., Wen, Z., Ashkan, A., & Szepesvari, C. (2015, February). Tight regret bounds for stochastic combinatorial semi-bandits."
> - "Gai, Y., Krishnamachari, B., and Jain, R. Combinatorial network optimization with unknown variables: Multi-armed bandits with linear rewards and individual observations."
>
> These foundational papers on CMAB focus on linear reward functions and semi-bandit feedback, making the inclusion of a cardinality constraint natural. In our case, $L \leq M$, and setting $L = M$ represents a special case of CMAB without cardinality restriction. Thus, by adding this restriction, our model becomes more general and better reflects real-world scenarios.
>
>
> ### 3. Question about algorithm CTS:
>
> We have noted the paper you mentioned that introduces the algorithm CTS. The reason we did not include this algorithm is that CTS is designed for CMAB with general reward functions and only achieves a suboptimal regret bound in the scenario of CMAB with linear reward functions.
>
> More specifically, the regret upper bound of CTS in our model is:
> $$
> O\left(\frac{ML^2}{\Delta_{S,\min} - 2(\ell^* + 2)\varepsilon}\log(T)\right),
> $$
> whereas the regret upper bound of CombUCB1, one of the benchmarks we use, is:
> $$
> O\left( \frac{ML^2}{\Delta_{S,\min}}\log(T) \right).
> $$
>
> Therefore, the best way to demonstrate the advantages of our algorithm is to compare it with CombUCB1 or MPMAB, which are more appropriate benchmarks in our context.
>
>
> ### 4. Question About the SD-CMAB Problem:
>
> Taking paths as an example, we can model each edge in a graph as a base arm,
> while the super arms represent paths formed by different combinations of these edges.
> The reward of a path reflects its performance, which may vary depending on specific contexts.
>
> Consider a scenario with four edges, denoted as $\{1,2,3,4\}$. If a particular property exists for edge 1—such as traversing edge 1 imposing an additional cost on subsequent edges—then we can group paths that include edge 1, such as $\{1,2\}$ or $\{1,3,4\}$, into a special class.
>
> The concept of a "class" in this context generalizes the "Set-Size Dependent" (SSD) model discussed in previous sections. While SSD-CMAB classifies paths solely based on their set size (i.e. the number of edges in one path), SD-CMAB allows for classification based on additional factors, such as properties or characteristics specific to individual edges.
>
> This broader framework enables a more flexible and detailed representation of combinatorial dependencies, extending the applicability of the model to more complex scenarios.

---

> ### Comment · Reviewer_8cZM · 2024-11-26
>
> Thank you for the answer.
> Though the denominators in the upper bound of the CTS and the CombUCB1 are slightly different, the orders are still the same. I believe the CTS has to be compared in the experiment section.
> Also, if the proposed algorithm can avoid computation heaviness by some interesting techniques, it should be in the main text, not in the appendix. This is because in combinatorial bandit, I believe computational efficiency is one of the most important aspects.

---

> > ### Author Response · Authors · 2024-11-27
> >
> > Thank you for your suggestions. We are currently doing experiments of CTS, and we would update the rebuttal revision if we have enough results before the deadline. We would also try to find some places in the main text to discuss the computation heaviness.

---

### Official Review · Reviewer_BjRj · 2024-11-04

**Soundness:** 3
**Presentation:** 4
**Contribution:** 3
**Rating:** 6
**Confidence:** 3

**Summary:**

The paper proposes a variant of the standard CMAB that allows arms to change reward means based on the set size of the super arm, which is pulled, provided that the *order* of the arms in terms of their reward means is preserved.

The paper tackles this new problem setup using a novel proposed algorithm SortUCB which splits the bandit process into three stages: Elimination, Sorting, and UCB Phase. SortUCB focuses on eliminating "bad" super arms fast and then focusing on a curated set of good super arms. The paper provides regret upper bounds as well as instance-dependent lower bounds on the set-size-dependent CMAB. The paper provides experimental synthetic evaluations on several baseline algorithms and showcases the improvement over them.

The paper then extends the current algorithm to set dependent combinatorial bandit where the base arms are allowed to have different reward distributions even with the same set size and the set of super arms can be dictated by further constraints. This broadly generalizes the initial setup and makes it much more applicable. They provide another algorithm SortUCB-SD to tackle this setup using a ($n_1, n_2$)-efficiency oracle and prove regret upper bound for the same.

**Strengths:**

The following would contribute to the strengths of the paper:
- **Clear Writing**: The paper is well written, precise, and to the point. The paper does a good job going slowly and explaining the inner workings of the algorithms and intuitive understanding of the paper

- **Innovative Algorithmic Solutions**: The novel proposed methods SortUCB and SortUCB-SD provide attractive regret upperbounds and their performance is further bolstered using the synthetic experiments.

- **Theoretical performance guarantees**: The paper provides theoretical proof of both the regret upper bound and the partial tightness to the fundamental lower bound for the SSD-CMAB problem.

- **Experiments**: The paper provides synthetic implementation with multiple baseline methods and showcases the prowess of the SortUCB method.

**Weaknesses:**

There are very few loopholes in the paper. The following are some points to work on further:

- **More justification on the Problem Setup**: The paper provides some justification on why this particular variant of CMAB is interesting and worth looking at, but it is not enough. A few dedicated paragraphs of potential application here would be a great addition.

- **Extension to broader class seems out of place**: The paper provides a great in-depth explanation and theoretical and experimental support to SortUCB, but the same is missing for SortUCB-SD.

- **A niche class of perfect lower and upper-bound matching**: The paper does attempt to explain in words how the regret bounds are tight. It might be better to phrase it as a corollary or a lemma on the small class of problems where it the upper and lower bound expressions are tight. Or is there a gap between the two fundamentals?

- **Experiment Replicability**: (Apologize if I missed this) I do not see any code files or URLs where I can run and verify the experimental evidence provided.

**Questions:**

Some questions for the authors :

- **Pivoting paper around SD-CMAD**: Observing that SD-CMAB is the more generic version of SSD-CMAB, why did the authors not decide to pivot the entire discussion around SD-CMAB and provide SSD-CMAB as a special case with better performance?

- **Real-world datasets**: Is it possible to have more real-world dataset experiments? this connects with requirements for more practical/ real-world problem setups similar to SSD-CMAB.

- **($n_1, n_2$)-efficiency oracle**: I can understand the need for such an oracle for comparison purposes. Is this a novel idea of the paper, or has this been used in essence in other literature? In either case, a thorough discussion on the same seems to be missing in the paper.

---

> ### Author Response · Authors · 2024-11-22
>
> We appreciate it for for your recognition of this paper and valuable advices.
>
> ### 1. Concern About Justification on the Problem Setup:
>
> Thank you for your suggestion. We have discussed some important motivations in the Introduction, specifically in paragraph 2. Due to space limitations, it may be challenging to include additional motivations or detailed applications in the current version of the paper. If we identify more appropriate motivations or examples in the future, we will make sure to include them in the paper as soon as possible.
>
> ### 2. Concern about broader class seems out of place:
>
> Thank you for the suggestion. We will keep doing experiments related to the SD-CMAB problem in the paper.
>
> ### 3. Concern About Perfect Lower and Upper Bound Matching:
>
> Thank you for the advice. Currently, we analyze the relationship between the upper bound and lower bound in both the "Contributions" section and in Remarks 3 & 4 following Theorem 2. In particular, Remark 4 demonstrates that when the minimum of   $\frac{\delta_{i(\ell)} - s_{i(\ell)}}{d_{\inf}(P_{i(\ell)}, \mu_{\ell^*(\ell)}, \mathcal{M}_{\ell})}$  for all $i \in [\ell^*+1, M]$ occurs at $L$, and $\ell^*$ is not approximately equal to $M$, the two bounds align with each other.
>
> Since the gap between the two bounds primarily depends on the differences between individual base arms,
> the gap itself is not the most significant factor in describing the advantage of our algorithms. Therefore, we analyze the relationship using remarks rather than introducing additional lemmas or corollaries.
>
> We will continue exploring more appropriate ways to express the tightness of the two bounds in future revisions.
>
>
> ### 4. Concern about Experiment Replicability:
>
> Thanks. We have added our code in the Rebuttal Revision.
>
> ### 5. Concern about pivoting paper around SD-CMAB:
>
> The reason why we don't pivot the discussion around SD-CMAB is that our current two main results are based on the SSD-CMAB problem, including the upper bound and lower bound. Hence, it is more suitable to focus on SSD-CMAB in the main paper and treat SD-CMAB as a generalization problem.
>
> We will continue researching the properties of the SD-CMAB problem to explore whether it can lead to more significant results in the future.
>
>
> ### 6. Concern about real-world datasets:
>
> Thank you for your suggestions. We will focus on the real-world dataset experiments in the future.
>
> ### 7. Detailed discussion of the Oracle:
>
> This Oracle is a novel contribution in our paper, which has not been included in previous CMAB studies. Due to space limitations, we have moved the detailed discussion of the Oracle to Appendix F.2. Please let us know if this section addresses your concerns effectively.

---

### Official Review · Reviewer_pong · 2024-11-06

**Soundness:** 2
**Presentation:** 3
**Contribution:** 3
**Rating:** 3
**Confidence:** 4

**Summary:**

This work proposes a new bandit setting,
called Set-Size Dependent Combinatorial Multi-Armed Bandits (SSD-CMAB). The key difference between SSD-CMAB with the classical CMAB is the reward distribution associated with each base arm depends on the size of the played super arm in that round.
They propose a novel algorithm, SortUCB, for solving SSD-CMAB.

The writing is not that clear. I suggest adding a learning protocol for your proposed learning problem. Also, using more than half a page for notations is not that space efficient. It can be simplified.

**Strengths:**

a new setting

**Weaknesses:**

The main concern is I am not convinced of the definition of regret. In my understanding, different rounds $t$ have different optimal arms, as the reward distributions depend on the size of the pulled super arm in that round. However, in (1), the optimal super arm is fixed.


Also, It is not fair to compare regret bounds for different settings, stated in the abstract.

Regret lower bound: partially tight? What does it not mean? Also, the derived regret lower bound is asymptotic. In addition, this work never specifies how to characterize a SSD-CMAB problem instance.

**Questions:**

Can SSD-CMAB be reduced to CMAB problem by setting some constraint to the super arms?

---

> ### Author Response · Authors · 2024-11-22
>
> Thank you for your invaluable review and advices.
>
> ### 1. For the concern about the space of notations:
> Thank you for your suggestions. We have simplified Section 2 by removing certain parts to address your concerns in the Rebuttal Revision. We appreciate your feedback and will continue revising the section to explore ways to streamline the content further while ensuring clarity and completeness.
>
>
> ### 2. For the concern about the definition of regret:
> Thank you for your concern regarding the definition of regret. We would like to clarify that the optimal arm is fixed in every bandit instance. Since we consider the stochastic setting with linear rewards, each base arm has a fixed reward expectation within super arms of the same set size. Given the action set of super arms, denoted by $\mathcal{S} = \{S \subseteq [M] \mid |S| \leq L\}$, the optimal arm can be identified by finding the super arm with the highest reward mean.
>
> For example, consider an SSD-CMAB instance with $M = L = 4$. Let the reward mean for base arm $i$ in a super arm of size $\ell$ be denoted by $\mu_{i(\ell)}$, where $i \in [M]$ and $\ell \in [L]$. The reward matrix for this instance is as follows:
>
> |          | $i=1$    |  $i=2$   |  $i=3$   |   $i=4$  |
> |----------|----------|----------|----------|----------|
> | $\ell=1$ | $\mu_{1(1)}=0.9$  | $\mu_{2(1)}=0.8$ | $\mu_{3(1)}=0.7$  | $\mu_{4(1)}=0.6$
> | $\ell=2$ | $\mu_{1(2)}=0.8$  | $\mu_{2(2)}=0.6$ | $\mu_{3(2)}=0.5$  | $\mu_{4(2)}=0.4$
> | $\ell=3$ | $\mu_{1(3)}=0.9$  | $\mu_{2(3)}=0.8$ | $\mu_{3(3)}=0.7$  | $\mu_{4(3)}=0.2$
> | $\ell=4$ | $\mu_{1(4)}=0.5$  | $\mu_{2(4)}=0.3$ | $\mu_{3(4)}=0.2$  | $\mu_{4(4)}=0.1$
>
> In this case, the reward mean of the super arm $S^* = \{1, 2, 3\}$ is higher than any other combination of base arms. Specifically, the reward mean of $S^*$ is
>
> $$r(S^*)=\mu_{1(3)}+\mu_{2(3)}+\mu_{3(3)}=0.9+0.8+0.7=2.4$$
>
> as the size of $S^*$ is $3$. This value is greater than the reward mean of any other super arm in this instance.
>
>
> ### 3. For the concern about comparisons of regret bounds:
> As explained in Section 4 of the Introduction, each SSD-CMAB instance can be viewed as a standard CMAB instance by treating each base arm in the super arms with varying set sizes as distinct base arms and independently learning their $ML$ distributions. Appendix B provides an algorithm to reduce any CMAB algorithm to handle the SSD-CMAB problem. However, standard CMAB algorithms do not account for the order-preservation property, preventing them from achieving optimal results. Consequently, the upper bound of any existing CMAB algorithm is inferior to ours (Theorem 1). This highlights the necessity of studying SSD-CMAB as a distinct research topic and developing algorithms tailored to its unique properties.
>
> ### 4. Addressing the Concern Regarding the "Partially Tight" Lower Bound:
>
> The term "partially tight" is clarified in Remarks 3 and 4 on page 7. Specifically, the upper bound in Theorem 1 is given as
> $$O\left( \max\left(\frac{M\delta_{L,\max}}{\Delta_{L,\min}^2}, \frac{L^2}{\Delta_{S,\min}}\right)\log(T) \right),$$
> while the lower bound in Theorem 2 can be informally written as
> $$\Omega\left(\max\left(\min_{\ell\in[L]}\left(\frac{(M-L)\delta_{\ell}}{\Delta_{\ell}^2}\right),\frac{L^2}{\Delta_{S,\min}}\right)\log(T)\right),$$
> as stated in the Abstract. It is evident that the second term in both the upper and lower bounds, i.e., $\frac{L^2}{\Delta_{S,\min}}\log(T)$, aligns, indicating that the regret caused by the UCB phase in Algorithm 1 is unavoidable. Meanwhile, the discrepancy in the first term between the upper and lower bounds can reduce to zero under specific conditions—namely, when the minimum in the first term of Theorem 2 consistently occurs at $L$ across $\ell \in [L]$, and $\ell^*$ is not approximately equal to $M$. For a detailed explanation of the term "partially tight," we direct reviewers to Remark 4.
>
>
> ### 5. For the Concern About SSD-CMAB Instance:
>
> Thank you for your suggestion. We have introduced the SSD-CMAB instance in Section 2, "THE SSD-CMAB PROBLEM," starting from the third paragraph of this section. We have also added a more detailed explanation of this term. Reviewers can refer to the "Rebuttal Revision" for the updated explanation.
>
>
> ### 6. For the Question of reduction to normal CMAB problem:
> Yes, it can be reduced to CMAB problem. We introduce the approach to consider each SSD-CMAB instance as a normal CMAB instance in the 4th paragraph of the Introduction section  (that is, treating each base arm in super arms with different set sizes as different base arms and independently learning $ML$ distributions).  The detailed implementation is provided in Appendix B.
>
> As stated in the Introduction section, using CMAB algorithms to solve the SSD-CMAB instance ignores the order-preservation property
> and results in a sub-optimal bound, which is worse than that of Algorithm 1. Therefore, it is crucial to conduct research specifically on the SSD-CMAB problem.

---

### Meta-Review · Area_Chair_awpb · 2024-12-21

**Metareview:**

This paper introduces a variant of the standard combinatorial multi-armed bandit (CMAB) problem, where the reward means of arms depending on the set size of the super arm being pulled, with the condition that the reward ranking of arms remains intact. The authors propose a novel algorithm, SortUCB, which divides the bandit process into three stages: Elimination, Sorting, and UCB Phase. SortUCB efficiently eliminates sub-optimal super arms and focuses on a curated set of promising ones. The paper provides regret upper bounds and instance-dependent lower bounds alongside experimental evaluations showing improvements over baseline algorithms.

The paper further extends the algorithm to a set-dependent combinatorial bandit problem, where base arms may have different reward distributions even with the same set size, and super arm sets can be constrained. This generalization increases the applicability of the setup. The authors propose the SortUCB-SD algorithm, which uses an efficiency oracle and proves a regret upper bound for this extended problem.

The paper suffers from some significant weaknesses: The definition of regret is unclear, as it assumes a fixed optimal super arm despite different arms being optimal in different rounds, and comparing regret bounds across different settings is not substantiated. The regret lower bound is only "partially tight," and key details about the SSD-CMAB problem instance are missing. The problem setup lacks sufficient real-world justification, and the extension to set-dependent combinatorial bandits is poorly supported both theoretically and experimentally. Algorithm 1 suffers from memory and computational complexity issues, and the lack of a time and space complexity analysis weakens the paper further. The work fails to adequately compare with closely related studies, particularly submodular bandits, and the empirical results are limited, with no updated experiments or code for replicability. The small dataset sizes and limited benchmarks make it difficult to justify the method's practicality, leading to the conclusion of not accepting the submission at this time. Based on the common consensus, therefore, I would recommend the authors submit to the next suitable venue addressing all the concerns once all the necessary modifications are incorporated.

**Additional Comments On Reviewer Discussion:**

See above.

---

### Decision · Program_Chairs · 2025-01-22

Reject